# PMIC: Improving Multi-Agent Reinforcement Learning with Progressive Mutual Information Collaboration

## Abstract

Learning to collaborate is critical in multi-agent reinforcement learning (MARL). A branch of previous works proposes to promote collaboration by maximizing the correlation of agents' behaviors, which is typically characterised by mutual information (MI) in different forms. However, simply maximizing the MI of agents' behaviors cannot guarantee achieving better collaboration because suboptimal collaboration can also lead to high MI. In this paper, we first propose a new collaboration criterion to evaluate collaboration from three perspectives, which arrives at a form of the mutual information between global state and joint policy. This bypasses the introduction of explicit additional input of policies and mitigates the scalability issue meanwhile. Moreover, to better leverage MI-based collaboration signals, we propose a novel MARL framework, called Progressive Mutual Information Collaboration (PMIC) which contains two main components. The first component is Dual Progressive Collaboration Buffer (DPCB) which separately stores superior and inferior trajectories in a progressive manner. The second component is Dual Mutual Information Estimator (DMIE), including two neural estimators of our new designed MI based on separate samples in DPCB. We then make use of the neural MI estimates to improve agents' policies: to maximize the MI lower bound associated with superior collaboration to facilitate better collaboration and to minimize the MI upper bound associated with inferior collaboration to avoid falling into local optimal. PMIC is general and can be combined with existing MARL algorithms. Experiments on a wide range of MARL benchmarks show the superior performance of PMIC compared with other MARL algorithms.

## 1 Introduction

With the potential to solve complex real-world problems, Multi-Agent Reinforcement Learning (MARL) has attracted much attention in recent years (Rashid et al., 2018; Iqbal & Sha, 2019; Wang et al., 2019b) and been applied to many practical domains like Game AI (Peng et al., 2017), robotics control (Matignon et al., 2012), transportation (Li et al., 2019). However, efficiently achieving collaboration and learning optimal policies still remains challenging in MARL (Liu et al., 2020; Wen et al., 2019). Prior efforts on improving collaboration of MARL agents mainly rely on efficient communication (Eccles et al., 2019; Das et al., 2019; Kim et al., 2019; 2021), or resort to value function factorisation (Sunehag et al., 2017; Rashid et al., 2018; Son et al., 2019).

A popular paradigm of cooperative MARL is Centralized Training, Decentralized Execution (CTDE) (Rashid et al., 2018; Sunehag et al., 2017). During centralized training, agents are granted access to other agents' information and possibly the global state, while during decentralized execution, agents make decisions independently based on their individual policies. However, although global information is incorporated during centralized training of CTDE, guiding the decentralized policies of multiple agents only by reward signals is often inefficient, especially when the reward signal is stochastic or sparse. This requires additional mechanisms to facilitate multi-agent collaboration (Roy et al., 2019; Kim et al., 2020). Later, another branch of works which is complementary to the above two branches focuses on leveraging the correlation or influence among agents for better collaboration (Jaques et al., 2018; 2019; Xie et al., 2020; Liu et al., 2020; Merhej & Chetouani, 2021).The intuition behind this is that if agents make decisions with the awareness of the influence

or behaviors of other agents, the non-stationary problem could be mitigated, thus agents are more likely to achieve collaboration.

Motivated by this, recently, several works (Chen et al., 2019; Mahajan et al., 2019; Kim et al., 2020) propose to maximize the correlation of agents' behaviors to promote collaboration. The correlation is commonly quantified by the mutual information (MI) of agents' behaviors. Signal Instructed Coordination (SIC) (Chen et al., 2019) incorporates a shared latent variable $z$ (sampled from a predefined distribution) into agents' policies as a signal of collaboration. During training, the MI between latent variable $z$ and agents' joint policy is maximized. By taking $z$ as input, the correlation of each agent's decision to the joint policy is enhanced. Similarly, Multi-agent Variational Exploration (Maven) (Mahajan et al., 2019) establishes an encoder network to encode the initial global state into the latent variable $z$ instead of sampling from a predefined distribution. Then Maven maximizes the MI between $z$ and the resulting trajectory of all agents. However, one common drawback of both methods is, the shared variable $z$ required during decentralized execution violates the CTDE paradigm. This makes algorithms fail in some real-world deployment scenarios where global communication is not available. Apart from the above methods, VM3-AC (Kim et al., 2020) measures the MI between any two agents' policies to capture the correlation of agents' behaviors, where $z$ is also introduced to guarantee the MI of any two policies is positive. Through maximizing the MI of any two agents' policies, VM3-AC reduces the uncertainty of agents' policies and thus facilitates multi-agent collaboration. However, VM3-AC also violates the CTDE paradigm due to the requirement of the shared variable $z$. Furthermore, VM3-AC calculates the MI between any two agents, which can be computationally infeasible with the increase of the number of agents. Another critical thing which is neglected by previous works is that, although agents with high degree of collaboration usually generate highly correlated behaviors, this does not necessarily results in high rewards. Thus, simply maximizing the MI of agents' behaviors cannot guarantee desired collaboration. Worse still, maximizing MI exacerbates the problem: the agent can easily overfit its strategy to the behaviours of other agents (Zhang et al., 2019; Lanctot et al., 2017). An example is shown in Figure 1 where agents are easily falling into sub-optimal collaboration.

In this paper, we propose a new collaboration criterion to evaluate the collaboration of multiple agents in terms of the joint policy, individual policy, and other agents' policies. Based on the criterion, we propose a new form of MI between the global state and joint policy to facilitate collaboration which is free of the reliance of explicit additional input as introduced in previous works and addresses the scalability issue. Moreover, to avoid falling into sub-optimal collaboration, we propose a novel framework, called *Progressive Mutual Information Collaboration* (PMIC), which contains two main components. The first component is *Dual Progressive Collaboration Buffer* (DPCB) which includes a positive and a negative buffer to dynamically keep the superior and inferior collaboration separately. The second component is *Dual Mutual Information Estimator* (DMIE) which has two MI neural estimators to estimate the MI of global state and joint action based on DPCB. Specifically, one estimator is trained on the positive buffer to provide the lower bound of MI, the other is trained on the negative buffer to provide the upper bound of MI. By maximizing the lower bound and minimizing the upper bound of MI, the agents could progressively break the current suboptimal collaboration and learn towards better performance which promotes an efficient and stable learning process. PMIC is general and can be easily combined with existing MARL algorithms. Our experiments show that PMIC significantly accelerates existing MARL algorithms, outperforms other related algorithms on a wide range of MARL benchmarks.

## 2 PRELIMINARIES

We consider a fully cooperative multi-agent task where a team of agents are situated in a stochastic, partially observable environment, it can be modeled as a *decentralised partially observable Markov decision process* (Dec-POMDP) (Oliehoek & Amato, 2016), defined by a tuple $\langle \mathcal{I}, \mathcal{S}, \mathcal{U}, \mathcal{O}, \mathcal{T}, \mathcal{R}, \gamma \rangle$. Here $\mathcal{I} = \{1, ..., N\}$ denotes the set of $N$ agents. In Dec-POMDP, the full state of the environment $s_t \in \mathcal{S}$ cannot be observed by agents at each time step $t$. Each agent $i \in I$ can only observe its individual observation $o_t^i$ determined by observation function $\mathcal{O}(s_t, i)$, each agent $i$ uses a stochastic policy $\pi_i$ to choose actions $u_t^i \sim \pi_i(\cdot|o_t^i)$, yielding the joint action $u_t = \{u_t^i\}_{i=1}^N \in \mathcal{U}$. After executing $u_t$ in state $s_t$, the environment transits to the next state $s_{t+1}$ according to transition function $\mathcal{T}(s_t, u_t)$ and agents receive a common reward $r_t$ from $\mathcal{R}(s_t, u_t)$, with a discount factor $\gamma$. In cooperative MARL, the collaborative team aims to find a joint pol-

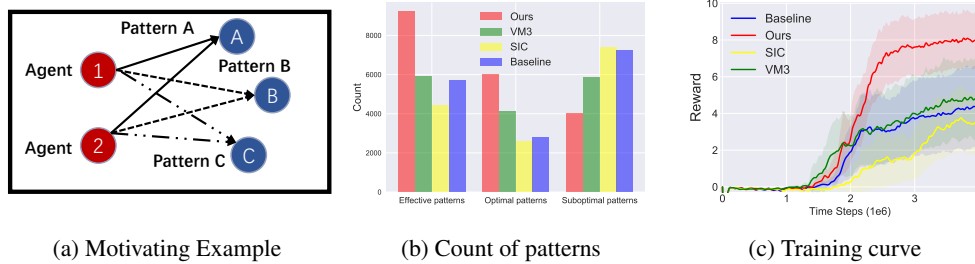

(a) Motivating Example      (b) Count of patterns      (c) Training curve

Figure 1: *(a)* A motivating environment. Effective patterns are patterns that can get positive rewards. The optimal pattern here is to achieve target A coordinately, while achieving other targets leads to sub-optimal patterns. *(b)* The count of different collaboration patterns learned by different algorithms, evaluated over 10k (1k x 10 seeds) episodes. *(c)* The training curves of episodic average rewards over 10 seeds with 95% confidence regions for different algorithms.

icy $\pi(u_t|s_t) = \prod_{a=1}^{N} \pi_i(u_t^i|o_t^i)$ that maximizes the total expected discounted cumulative reward $\mathbb{E}_\pi\left[\sum_{t=0}^{\infty} \gamma^t r_t\right]$ where $y$ is the discount factor.

# 3 PROGRESSIVE MUTUAL INFORMATION COLLABORATION

## 3.1 MOTIVATION

In this section, we show the motivation of our work to explain why blindly maximizing correlation of agents' behaviors may make agents stuck into suboptimal collaboration. Firstly, we give a motivating example shown in (a) of Figure 1 where agents need cooperate to catch the targets A, B and C with different sparse rewards and punishments. Capturing A is more rewarding than B and C. Agents may receive penalties for capturing different targets. We denote the behaviors generated by the joint policy of achieving a certain target concurrently as *collaboration pattern*. In this setting, promoting

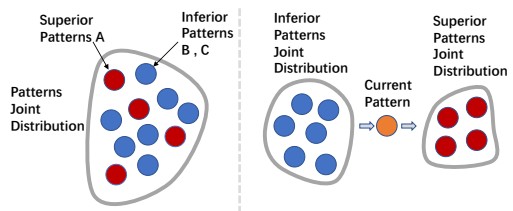

Figure 2: The left represents maximizing MI on all patterns and the right represents maximizing MI with superior patterns and minimizing MI with inferior patterns.

collaboration with simply maximizing the MI of agents' behaviors cannot guarantee a better collaboration. As shown in Figure 2, the *patterns joint distribution* is composed of patterns which capture targets A, B and C. This distribution is chaotic and contains various collaboration patterns. In this case, maximizing MI on any patterns of the distribution may enhance the inferior collaboration patterns, for example maximizing MI on the pattern B or C may make agents converge to a sub-optimal collaboration pattern. In fact, this is a very common situation. In complex environments, there is a wide variety of collaboration, maximizing MI could only stimulate agents towards a certain collaboration, which may be sub-optimal and in turn prevents from discovering better collaborations. This happens because, as learning preceeds, a seemly good collaboration will become sub-optimal once a better one is discovered. However, if we excessively maximize the MI using sub-optimal samples, the agents may be accelerated to fall into sub-optimal policies. Thus the motivation of our method is to solve the limitation with the maximization-minimization MI. As shown in the right of Figure 2, we divide the distribution into the superior and inferior distributions. then we maximize the MI associated with superior collaboration patterns and minimize the MI associated with inferior collaboration patterns. In this manner, the current patterns strengthen towards superior patterns and are away from inferior patterns.

Next we conduct experiments with four methods on the environment: MADDPG (Lowe et al., 2017), SIC-MADDPG (Chen et al., 2019), VM3-MADDPG (Kim et al., 2020) and our method where VM3 and SIC both promote collaboration by maximizing MI. Results of the count of algorithms converging to different collaboration patterns and overall performance are shown in (b) and (c) of Figure 1. We can see that both SIC and VM3 have a greater probability of falling into the sub-

optimal patterns than that into the optimal one. This demonstrates that maximizing MI only cannot guarantee the optimal collaboration, it may also stuck into sub-optimal collaboration. In addition, our method outperforms other methods in both the number of effective patterns and the optimal patterns, which demonstrates that our method is more effective in facilitating collaboration while avoiding falling into sub-optimal collaboration patterns than other methods. In addition to the above problem, VM3-AC (Kim et al., 2020) has scalability issue which need to compute the MI of policies between any two agents. Besides, all previous methods (Kim et al., 2020; Chen et al., 2019; Mahajan et al., 2019) need additional inputs, e.g., latent variable $z$ in (Chen et al., 2019; Kim et al., 2020) or global initial state in (Mahajan et al., 2019), during the decentralized execution phase, which violates the rule of CTDE paradigm.

To this end, we propose a new collaboration criterion as well as a novel framework to solve the above problems. Specifically, we propose a new collaboration criterion to evaluate the collaboration of multiple agents in terms of the joint policy, individual policy, and other agents' policies. Based on the criterion, a new way that calculates the MI of the global state and joint policy is proposed to promote collaboration which does not require an additional latent variable as input and addresses the scalability issue. To solve the problem of easily falling into sub-optimal collaboration by maximizing MI only, we then propose PMIC framework which leverages two distinct mechanisms to guide agents in a progressive manner. Finally, to solve the inefficiency of being guided only by rewards (e.g., stochastic or sparse), we use the estimated values of the new MI which are provided by DMIE as intrinsic rewards. To the best of our knowledge, we are the first to propose a combination of maximizing the lower bound of MI and minimizing the upper bound of MI to promote collaboration in MARL. In the following section, we first introduce the new collaboration criterion, then we show the details of PMIC framework, finally we give an combination of our method with MADDPG.

## 3.2 COLLABORATION CRITERION

In this section, we propose a new collaboration criterion to evaluate the collaboration quality in multi-agent systems. We suppose a good collaboration should contain the following three properties : 1) the joint policy should explore in the joint action space to enrich the diversity of the collaboration patterns and avoid model collapse. 2) Each agent $i$ should be deterministic enough about what behaviours it should adopt given its observation. 3) Meanwhile, given an agent $i$'s policy and other agents' observation, the uncertainty on other agents' policies should be low, thus more easily to achieve better collaboration. Based on this, the new learning objective considers the above three aspects: 1) maximizing $H(\pi(\cdot|s))$; 2) minimizing the entropy of each agent's individual policy: $H(\pi_i \mid s)$, where $\pi_i$ is the policy of agent $i$; 3) minimizing each agent's uncertainty on other agents' policies: $H(\pi_{-i} \mid \pi_i, s)$, where $-i$ is other agents except agent $i$ and $\pi_{-i}$ is the joint policy of other agents. Thus our final objective can be written as:

$$H(\pi(\cdot|s)) - H(\pi_i \mid s) - H(\pi_{-i} \mid \pi_i, s) = H(\pi(\cdot|s)) - H(\pi \mid s) = I(s; \pi(\cdot|s)). \quad (1)$$

The objective ultimately can be defined as the mutual information of the global state and joint policy. Thus we can maximize the MI of global state and joint policy to promote collaboration. This MI objective is simple and general that does not rely on any latent variable or the scalability issue. However, simply maximizing this MI can hinder algorithm learning as discussed above. Therefore, we propose a novel framework PMIC to solve the problem, which is detailed in the following section.

## 3.3 PMIC FRAMEWORK

In this section, we introduce our framework, Progressive Mutual Information Collaboration (PMIC), to leverage the collaboration criterion proposed above to facilitate multi-agent collaboration. The key idea of PMIC framework is to identify superior and inferior collaboration patterns, and encourage multiple agents to achieve superior patterns while prevent agents from performing inferior patterns. Such a guidance of joint policy is imposed in a progressive manner during the learning process, since collaboration patterns are not known in advance with an oracle view. Intuitively, with the help of PMIC, agents alter their collaboration patterns progressively towards better ones while avoid being trapped at sub-optimal collaboration patterns.

To be specific, Figure 3 illustrates PMIC framework, consisting of two main components. The first component is Dual Progressive Collaboration Buffer (DPCB) which separately stores superior and inferior trajectories progressively. DPCB maintains historical samples that reflect superior

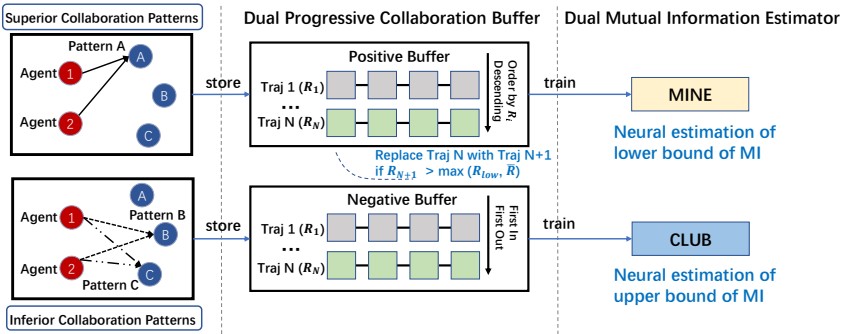

Figure 3: An overall illustration of PMIC framework, consisting of two main components. Dual Progressive Collaboration Buffer (DPCB) maintains superior and inferior trajectories separately in a progressive manner. Dual Mutual Information Estimator (DMIE) includes two neural estimators of the collaboration criterion (Equation 1) based on separate samples in DPCB.

and inferior patterns progressively. The second component is Dual Mutual Information Estimator (DMIE), including two neural estimators of our MI collaboration criterion based on separate samples in DPCB. DMIE provides quantitative measurements of state and joint policy variables based on their samples, which is used to improve agents' policies by maximizing the MI estimates associated with superior collaboration pattern and minimizing the MI estimates associated with inferior ones. In the following, we introduce the two components in details.

**Dual Progressive Collaboration Buffer (DPCB).** DPCB consists of a positive buffer and a negative buffer to store superior and inferior trajectories respectively. To identify the superior trajectories, we use the average return $\bar{R}$ over the last $M$ episodes as a baseline. We denote the trajectory with the lowest return in the positive buffer as $R_{\text{low}}$. For episode $k$, we store the trajectory with return $R_k$ if $R_k > \max(R_{\text{low}}, \bar{R})$; when the positive buffer is full, DPCB overwrites the trajectories with return $R_{\text{low}}$ is overwritten. This ensures the quality of collaboration patterns reflected by the positive buffer monotonically increases during the learning process. For the opposite case $R_k \leq \max(R_{\text{low}}, \bar{R})$, the trajectory is stored in the negative buffer, which is maintained in a First-In-First-Out (FIFO) manner. This is to make agents break recent inferior collaboration patterns, thus driving agents to adjust and explore consistently. The positive buffer and negative buffer store the samples that reflect superior and inferior collaboration patterns encountered during the learning process.

**Dual Mutual Information Estimator (DMIE).** Based on the samples in DPCB, quantitative estimation of collaboration criterion proposed in Sec.3.2 is obtained by training neural estimators of $I(s; \pi(\cdot|s))$. DMIE consists of a MINE (Belghazi et al., 2018) estimator of the lower bound of $I(s; \pi(\cdot|s))$ for maximization based on the positive buffer, and a CLUB (Cheng et al., 2020) estimator of the upper bound of $I(s; \pi(\cdot|s))$ for minimization based on the negative buffer. In concrete, MINE approximates the lower bound of $I(s; \pi(\cdot|s))$ below, based on states $s$ and joint action samples $u \sim \pi(\cdot|s)$:

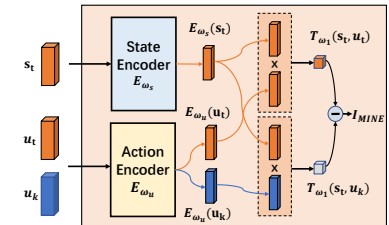

Figure 4: Architecture of MINE for lower-bound estimation of $I_{\text{MINE}}(s; u \sim \pi(\cdot|s))$. $(s_t, u_t) \sim \mathbb{P}_{\mathcal{SU}}$ and $(s_t, u_k) \sim \mathbb{P}_{\mathcal{S}} \otimes \mathbb{P}_{\mathcal{U}}$.

$$I(s; \pi(\cdot|s)) \geq I_{\text{MINE}}(s; \pi(\cdot|s)) = \sup_{\omega_1 \in \Omega} \underbrace{\mathbb{E}_{\mathbb{P}_{\mathcal{SU}}}\left[-sp\left(-T_{\omega_1}(s, u)\right)\right] - \mathbb{E}_{\mathbb{P}_{\mathcal{S}} \otimes \mathbb{P}_{\mathcal{U}}}\left[sp\left(T_{\omega_1}(s, u)\right)\right]}_{-\mathcal{L}(\omega_1)}, \quad (2)$$

where $\mathbb{P}_{\mathcal{SU}}$ is state-action joint distribution $s \sim D_s(\cdot)$ and $u \sim \pi(\cdot|s)$ with some buffer $D$; and $\mathbb{P}_{\mathcal{S}}$ and $\mathbb{P}_{\mathcal{U}}$ are the marginal distribution of $\mathbb{P}_{\mathcal{SU}}$, where the samples can be obtained by $s \sim D_s(\cdot)$ and $u \sim D_u(\cdot)$. $T_{\omega_1}(s, u)$ is a neural network with parameters $\omega_1 \in \Omega$ that outputs a scalar and $sp(z) = \log(1 + \exp(z))$. Similarly, CLUB approximates the upper bound of $I(s; \pi(\cdot|s))$:

$$I(s; \pi(\cdot|s)) \leq I_{\text{CLUB}}(s; \pi(\cdot|s)) = \underbrace{\mathbb{E}_{\mathbb{P}_{\mathcal{SU}}}\left[\log T_{\omega_2}(u \mid s)\right] - \mathbb{E}_{\mathbb{P}_{\mathcal{S}} \otimes \mathbb{P}_{\mathcal{U}}}\left[\log T_{\omega_2}(u \mid s)\right]}_{-\mathcal{L}(\omega_2)}, \quad (3)$$

where $T_{\omega_2}(u|s)$ is a neural network with parameters $\omega_2$ that models the conditional distribution.

The training losses of MINE and CLUB neural estimators are $\mathcal{L}(\omega_1)$ and $\mathcal{L}(\omega_2)$ defined in above equations. Thus the total loss of DMIE is:

$$\mathcal{L}_{\text{DMIE}} = \mathcal{L}(\omega_1) + \mathcal{L}(\omega_2). \qquad (4)$$

For calculation, the expectation of state-action joint distribution is replaced by sampling $s \sim D_s(\cdot)$ and $u \sim \pi(\cdot|s)$ from the positive buffer and negative buffer respectively. Detailed architecture of MINE and CLUB are illustrated in Figure 4 and 5. We provide a complete process of training and calculation in

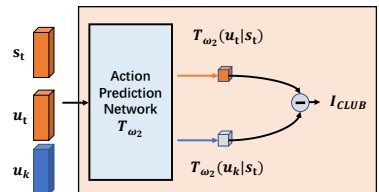

Figure 5: Architecture of CLUB upper-bound estimation of $I_{\text{CLUB}}(s; u \sim \pi(\cdot|s))$. $T_{\omega_2}(u_t|s_t)$ and $T_{\omega_2}(u_k|s_t)$ are the probabilities of inferring $u_t$ and $u_k$ based on $s_t$.

Appendix C. After training, given state and joint-action samples, we can use MINE and CLUB neural estimators to calculate the MI estimates according to Equation 2 and 3.

MINE and CLUB are trained with the samples from the positive buffer and negative buffer respectively. Intuitively, only the behaviors that obey superior collaboration patterns or inferior collaboration patterns have large MI estimates calculated by MINE or CLUB, since $\mathcal{L}(\omega_1)$ and $\mathcal{L}(\omega_2)$ are optimized. Therefore, given the interaction samples of current joint policy of agents, MINE and CLUB can calculate the MI estimates which provide effective signals in guiding agents' behavior towards or away from superior and inferior collaboration patterns progressively. We introduce how to integrate PMIC framework with MARL for more efficient learning in the following.

### 3.4 INTEGRATION OF PMIC AND MARL

Recall the key idea of PMIC framework: to leverage superior and inferior collaboration patterns encountered during the learning process to provide effective learning signals that encourage agents to achieve better collaboration progressively while avoid being trapped by sub-optimal ones. With the quantitative MI estimation as introduced in previous subsection, we aim to find the joint policy that maximizes the MI estimate from MINE and minimizes the MI estimate from CLUB in addition to maximizing expected discounted return conventionally. Formally, the objective function of PMIC-MARL is: $J(\pi) = \mathbb{E}_\pi \left[ \sum_{t=0}^\infty \gamma^t r_t \right] + I_{\text{MINE}}(s; \pi(\cdot|s)) - I_{\text{CLUB}}(s; \pi(\cdot|s))$. Note all three terms to optimize in the equation are expectation over trajectories generated by policy $\pi$. We then transform the two MI estimation terms into fractions of each time step and arrive at the following objective with per-step MI rewards:

$$J(\pi) = \mathbb{E}_{s,u\sim\pi} \left[ \sum_{t=0}^\infty \gamma^t (r_t + r_t^{\text{DMIE}}) \right], \text{ where } r_t^{\text{DMIE}} = \alpha I_{\text{MINE}}(s_t; u_t) - \beta I_{\text{CLUB}}(s_t; u_t). \qquad (5)$$

$\alpha$ and $\beta$ are the hyperparameters that control the dependence on MI guidance. In principle, PMIC-MARL is a general framework which can be implemented with different MARL algorithms.

For a representative implementation, we introduce PMIC-MADDPG by integrating PMIC framework with MADDPG (Lowe et al., 2017). Other PMIC-MARL implementation is also studied in our experiments. The pseudo-code of PMIC-MADDPG is shown in Algorithm 1. In each episode, agents interact with the environment and store the trajectory samples into experience replay $D$ (Line 3-7 in Algorithm 1). The trajectory is added to the positive buffer or the negative buffers in DPCB according to its return (Line 8 in Algorithm 1). In a frequency of $k$ steps, DMIE is trained with the samples from DPCB (Line 10-11 in Algorithm 1) to consistently reflect current superior and inferior collaboration patterns. At last, PMIC-MADDPG updates Q networks and the policies of agents based on DMIE (line 12 in Algorithm 1), $\mathcal{L}_Q(\phi)$ and $\mathcal{L}_\pi(\theta)$ are the objective functions of Q and policy which are defined as:

$$\mathcal{L}_Q(\phi) = \mathbb{E}_{s,u,r,s'\sim\mathcal{D}} \left[ (\hat{y} - Q_\phi(s,u))^2 \right]; \mathcal{L}_\pi(\theta) = \mathbb{E}_{s\sim\mathcal{D}}[-Q_\phi(s, \pi_\theta(u|s))], \qquad (6)$$

where $\hat{y} = r + r^{\text{DIME}} + \gamma Q_{\phi'}(s', \pi_\theta'(s'))$. With PMIC, the agents can progressively break the current sub-optimal behavioral patterns and learn towards better collaboration, which promotes an efficient and stable learning process.

### 4 EXPERIMENTS

In this section, we design experiments to answer the following questions:

---

**Algorithm 1** PMIC-MADDPG

---

1: **Initialize:** Q networks $\phi$, N actor networks $\theta_1...\theta_n$ and corresponding target networks $\phi'$, $\theta_1'$, ..., $\theta_n'$; DMIE with parameters $\omega_1$ and $\omega_2$ and DMIE update frequency $k$; DPCB; experience replay buffer $\mathcal{D}$
2: **for** each episode **do**
3:     **for** each time steps $t$ **do**
4:         **for** each agents $i$ **do**
5:             Select action $u_t^i \sim \pi_{\theta_i}(o_t^i)$ for each agent $i$
6:         Execute joint action $u_t$ and observe observations $\{o_{t+1}^i\}_{i=1}^n$ (or global state $s_{t+1}$), reward $r_t$
7:         Store interaction trajectories to $D$
8:         Add interaction trajectories to the positive buffer or the negative buffer in DPCB (according Sec.3.3)
9:         **for** each training step $m$ **do**
10:             **if** $m\%k == 0$ **then**
11:                 Update DMIE with DPCB according to Equation 4
12:                 Update Q and actor networks according to Equation 6

---

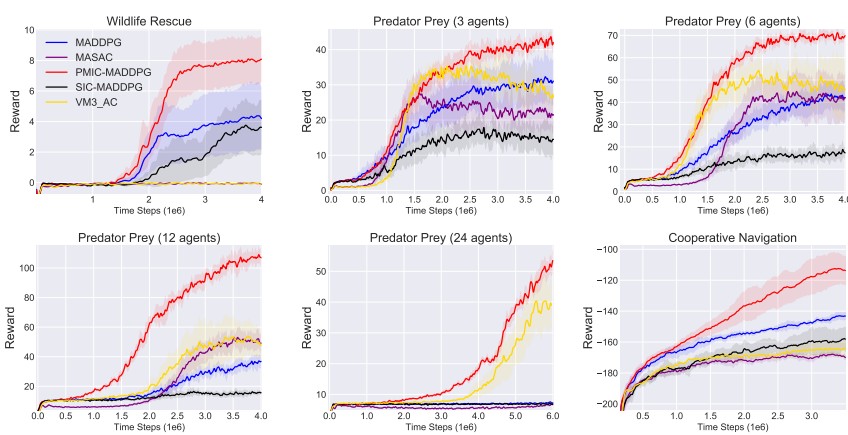

Figure 6: Comparisons of averaged return on MPE.

**RQ1 (Performance)** Can PMIC effectively improve algorithms and outperform other methods?
**RQ2 (Effectiveness of maximizing and minimizing $I(s; \pi(u|s))$)** Whether $I(s, \pi(u|s))$ is effective? can both mechanisms bring performance improvements?
**RQ3 (Effectiveness of DPCB)** Is DPCB a necessary component for PMIC?

**Benchmark & Baselines**. For a comprehensive comparative study, we evaluate our algorithms on both discrete and continuous action spaces. For the continuous action space, two kinds of domains are considered: Multi-Agent Particle Environment (MPE) and Multi-Agent MuJoCo benchmark, and we compare PMIC-MADDPG with six advanced algorithms as baselines: SIC-MADDPG (Chen et al., 2019), VM3-AC (Kim et al., 2020), MASAC (Kim et al., 2020), MADDPG (Lowe et al., 2017), FacMADDPG (de Witt et al., 2020), COMIX (de Witt et al., 2020) where FacMADDPG and COMIX are the state-of-the-art (SOTA) algorithms in Multi-Agent MuJoCo benchmark (de Witt et al., 2020). For environments in discrete action space, the challenging StarCraft II micromanagement (SMAC) benchmark is considered which has high complexity of control and requires learning policies in a large discrete action space, and we compare PMIC-RODE with the current SOTA algorithm RODE on SMAC. Environment description is provided in Appendix B.

For all baseline algorithms, we use code provided by their authors or build the algorithm according to the original papers where the hyperparameters have been fine-tuned on all environments. Further implementation details can be found in Appendix H.

## 4.1 PERFORMANCE (RQ1)

We first evaluate the performance on 6 environments of MPE with 10 different random seeds: Wildlife Rescue, Cooperative navigation, Partial Observation Cooperative Predator Prey with 3, 6, 12, and 24 predators respectively, where the agents control predators for collaboration and the policy of prey is fixed. The results are shown in Figure 6. PMIC-MADDPG has a significant improvement

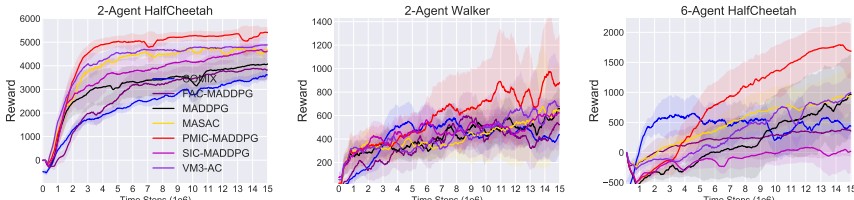

Figure 7: Comparisons of averaged return on Multi-Agent MuJoCo.

for MADDPG and outperforms other methods across all tasks. On these environments, a higher reward means a better collaboration pattern. We can see that both VM3-AC and SIC-MADDPG get lower rewards than PMIC-MADDPG which indicates maximizing MI makes agents fall into sub-optimal collaboration pattern quickly and cannot guarantee a better collaboration pattern. Besides, PMIC can still deliver significant performance gains as the number of agents becomes larger, while some other algorithms cannot learn to collaborate effectively. For example, on Predator Prey with 24 agents, only PMIC-MADDPG and VM3-AC can learn collaborative behaviors, where VM3-AC requires three times more time consumption than PMIC-MADDPG.

We further evaluate PMIC on 3 tasks of Multi-Agent MuJoCo benchmark with 10 different random seeds. In these tasks, agents need to cooperate in robot control and different agents control different joints of the robot. In our experimental setting, agents do not share information, which is the most difficult setting in Multi-Agent MuJoCo. The experimental results in Figure 7 show that PMIC-MADDPG performs significantly better than MADDPG on the tasks. On the other hand, PMIC-MADDPG outperform other baselines (COMIX,Fac-MADPPG,SIC-MADDPG,VM3AC,MASAC), which demonstrates the benefits of PMIC for improving performance in challenging cooperative tasks with partial observation and continuous action space.

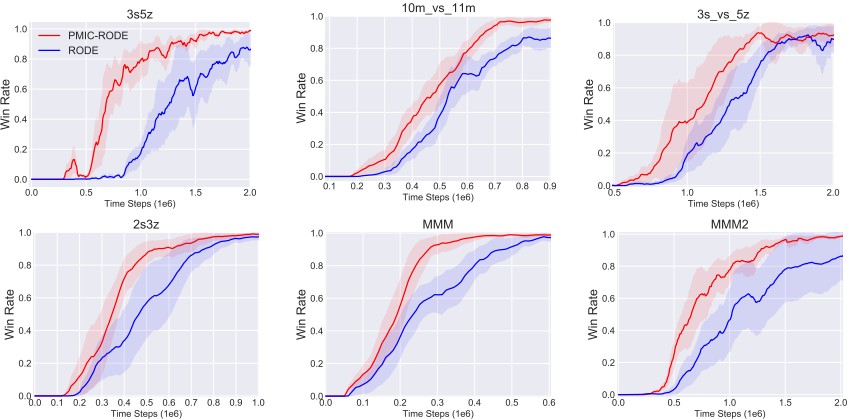

Figure 8: Comparisons of averaged test win rate on SMAC.

For SMAC, we evaluate PMIC on 6 maps with 5 different random seeds. Note that RODE is the SOTA algorithm on SMAC thus we mainly compare our method with RODE. The results are shown in Figure 8, PMIC can also provide an improvement for RODE with faster rate of convergence and higher performance. In summary, we show that PMIC is an effective framework that can be integrated with most algorithms and can provide significant improvements both in continuous action space and discrete action space. More experiments are included in Appendix F.

## 4.2 Effectiveness of maximizing and minimizing $I(s; \pi(u|s))$ (RQ2)

To answer RQ2, we first provide an ablation study on maximizing and minimizing $I(s; \pi(u|s))$ to investigate the effect of the two mechanisms. As shown in (a) and (b) of Figure 9, using maximizing and minimizing $I(s; \pi(u|s))$ can achieve faster convergence and higher performance. than only using maximizing or minimizing $I(s; \pi(u|s))$.

To verify whether the first estimator MINE that maximizes $I(s; \pi(u|s))$ can guide agents correctly, we collect optimal and sub-optimal collaboration patterns to train MINE respectively, then use the

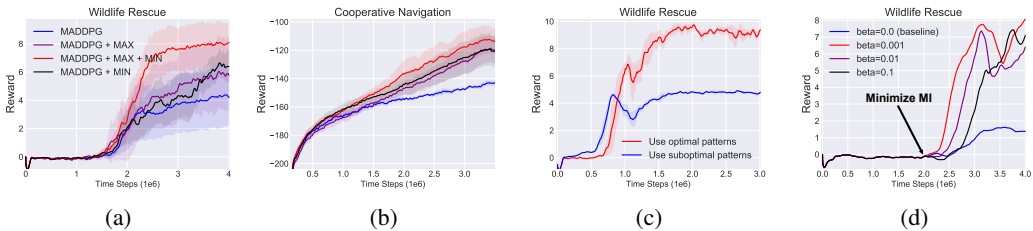

Figure 9: Ablation study and effect about maximizing and minimizing $I(s; \pi(u|s))$.

trained MINE to guide an initialized MADDPG from the beginning. As shown in (c) of Figure 9, training MINE with optimal collaboration patterns can quickly guide the algorithm to learn the optimal patterns, which indicates that maximizing $I(s; \pi(u|s))$ on superior collaboration patterns can guide agents correctly and quickly. However, with the guidance of MINE trained on sub-optimal patterns, the policy only converges to the sub-optimal patterns. This indicates that only maximizing MI without distinguishing the quality of collaboration patterns will make agents fall into inferior collaboration patterns. This further verifies our assumption in the motivation section and also indicates the necessity of designing DPCB.

Next we examine whether the second estimator CLUB that minimizes $I(s; \pi(u|s))$ can break the current sub-optimal collaboration pattern and help agents achieve better collaboration. We test on Wildlife Rescue and find a policy which has trapped into suboptimal patterns(e.q., rescue animals with lower rewards). After 2 million time steps, we add minimizing $I(s; \pi(u|s))$ for the policy with different $\beta$. The result in (c) of Figure 9 shows that the training curve gradually increases and achieves higher rewards than baseline. This indicates that minimizing $I(s; \pi(u|s))$ can help the algorithm break the current sub-optimal pattern and discover better collaboration patterns. Due to space limitation, more ablation experiments are putted in Appendix F.

### 4.3 EFFECTIVENESS OF DPCB (RQ3)

We finally examine whether the positive buffer and the negative buffer in DPCB can be replaced with a normal replay buffer. The results are shown in Figure 10. Leveraging DPCB is the most effective way. In fact, leveraging normal replay buffer can cause patterns conflicts which means training MINE and CLUB with same patterns and can not guarantee the correct guidance.

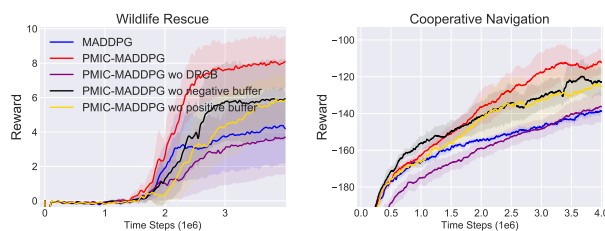

Figure 10: Ablation experiments on DPCB.

## 5 CONCLUSION

In this paper, we propose a novel collaboration criterion in a form of the mutual information between global state and joint policy. This bypasses the introduction of explicit additional input of policies that violates CTDE paradigm and mitigates the scalability issue meanwhile. Moreover, to address the potentially detrimental encouragement induced by mutual information maximization, we proposes a framework called PMIC, consisting of: DPCB which progressively maintains superior and inferior collaboration patterns; and DMIE which has two MI neural estimators of our MI collaboration criterion based on separate samples in DPCB. In additional to maximizing expected discounted return, PMIC-MARL aims at maximizing the MI estimates associated with superior collaboration to guide agents to facilitate better collaboration and minimizing the MI estimates associated with inferior collaboration to encourage exploration and avoid falling into sub-optimal collaboration. In our experiments, we evaluate several implementations of PMIC-MARL in a wide range of cooperative environments with both continuous action space and discrete action space. The results demonstrate that the effectiveness and generalization of PMIC.

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

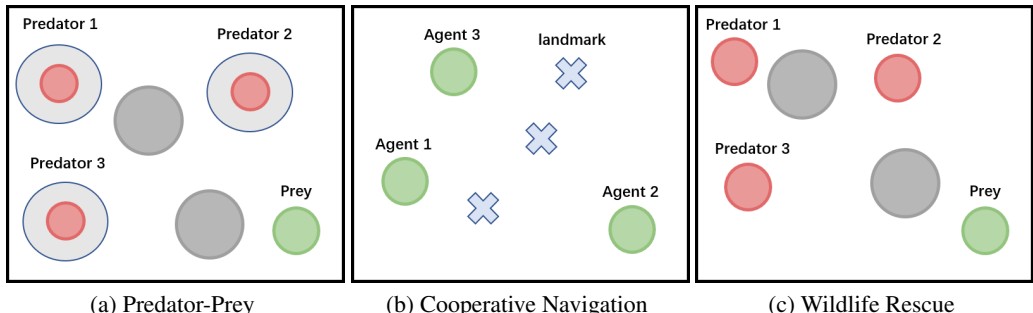

(a) Predator-Prey      (b) Cooperative Navigation      (c) Wildlife Rescue

Figure 11: Multi-Agent Particle Environment.

## A  RELATED WORK

We have divided previous works on facilitating multiple agents collaboration into three branches in Sec.1. In this paper, we mainly pay attention to the third branch, where the previous works promote agents collaboration by increasing the correlation of agents. We divide the previous works in two categories. In the first category, the algorithms take a holistic view and facilitate collaboration by increasing the correlation (i.e., mutual information) of the agent' own behaviours and the joint policy. For example, SIC (Chen et al., 2019) extracts the information of the joint policy into the latent variables, which are then used as the input of agents' policy networks, then SIC maximizes mutual information of each agent own behaviors and the latent variables to improve the correlation of agents. Based on SIC, the agents will know what kind of the joint policy the whole team is taking and what kind of action should be selected based on the latent variables. MAVEN (Mahajan et al., 2019) has a similar idea with SIC, except that MAVEN extracts joint policy information from the initial global state and maximizes the mutual information of future trajectories and the latent variables. In the second category, the algorithms facilitate collaboration from the perspective of individual agents by enhancing the correlation among different agents. For example, EITI (Wang et al., 2019a) leverage MI to captures the influence between an agent's current actions/states and other agents' next states in grid environments. SI-MOA (Jaques et al., 2019) proposes a social influence intrinsic reward to captures the mutual information between multiple agents' actions to achieve coordination. VM3 (Kim et al., 2020) has a similar idea with SIC-MOA except that VM3 modifies policy iteration based the MI and introduces additional input to explicitly represent the relation of agents' policies. Finally, VM3 achieves better performance than SIC-MOA. In this paper, we propose a new collaboration measurement by mutual information between the global state and the joint policy. Besides, our method consider the problem introduced by maximizing MI and propose the maximization-minimization MI to solve the problem of falling into inferior collaboration patterns.

## B  ENVIRONMENT DETAILS

**Predator Prey**: N slower cooperating agents chase the faster adversary around a randomly generated environment with L large landmarks impeding the way. In our setting, agents control the predators to chase the prey, the policy of prey is fixed. The agents are partially observable, the observation radius of each predator is 0.25. Only when the predator captures prey, predator can get the reward 10. we set N to 3, 6, 12, 24 separately. Each game has 25 steps.

**Cooperative Navigation**: Agents must cooperate through physical actions to reach a set of L landmarks. In our setting, agents receive shared reward which is the sum of the minimum distance of the landmarks from any agents, and the agents who collide each other receive negative reward -1. Besides, all agents receive 1 if all landmarks are occupied. Each game has 25 steps.

**Wildlife Rescue**: N agents must cooperate to rescue M wildlife with different risks and rewards. We provide a control time $T_c$. When an agent catches up with an animal, the agent can control $T_c$

seconds to wait for other agents to arrive. Each game has $T$ steps. We set $T_c$ to 8, $T$ to 60. The specific reward at the end of each episode is set in Table 1.

Table 1: The reward matrix of the rescue agents at the end of each episode. Both agents receive the same reward.

| reward    agent 2 
 agent 1 | tiger | deer | cat | on the road |
|---|---|---|---|---|
| tiger | 11 | -30 | 0 | -30 |
| deer | -30 | 7 | 6 | -30 |
| cat | 0 | 6 | 5 | 0 |
| on the road | -30 | -10 | 0 | 0 |

Multi-Agent MuJoCo (de Witt et al., 2020) is a range of challenging continuous multi-agent control tasks of Multi-Agent MuJoCo benchmark suite. Multi-Agent MuJoCo is designed for decentralized cooperative continuous multi-agent robotic control. The structure is shown in Figure 12. For evaluation, we use the reward setting of the original paper but we set $k$ to zero for all environments. $k$ controls how much information each agent can observe from its adjacent agents. When $k$ is zero, it means each agent can only observe information about its own joints, which is the hardest setting for Multi-Agent MuJoCo.

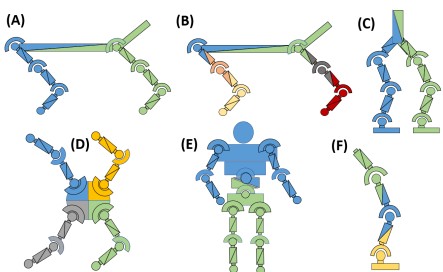

Figure 12: The structure of Multi-Agent MuJoCo. Different colored parts of the robot are controlled by different agents.

For SMAC, we use the latest version 2.4.10 and default settings for all maps.

## C  PROCESS OF TRAINING MINE AND CLUB

In this section, we detail the process of training MINE and CLUB.

First we introduce the process of training MINE. We train MINE based on the positive buffer of DPCB. Figure 4 illustrates the detailed process of estimating the lower bound. We design the network $T$ with parameters $\omega_1$ as a state encoder $E_{\omega_s}$ and an action encoder $E_{\omega_u}$. MINE takes the $(s_t, u_t) \sim \mathbb{P}_{\mathcal{SU}}$ and $(s_t, u_k) \sim \mathbb{P}_{\mathcal{S}} \otimes \mathbb{P}_{\mathcal{U}}$ as input and encodes the data as vectors $E_{\omega_s}(s_t)$, $E_{\omega_u}(u_t)$, $E_{\omega_u}(u_k)$ with the same dimension. Then we take the inner product of $E_{\omega_s}(s_t)$ and $E_{\omega_u}(u_t)$, $E_{\omega_s}(s_t)$ and $E_{\omega_u}(u_k)$ to get $T_{\omega_1}(s_t, u_t)$ and $T_{\omega_1}(s_t, u_k)$ (e.q., scores) separately. Finally we can plug the scores into Equation (2) to the estimate of $I_{MINE}(s_t, u_t)$. In summary, MINE is trained by sampling data from positive buffer of DPCB to minimize $\mathcal{L}(\omega_1)$ in Equation (2).

Then we details the process of training CLUB. We train CLUB based on the negative buffer of DPCB. Figure 5 illustrates the detailed process of estimating the upper bound. We design the network $T$ with parameters $\omega_2$ as action prediction network. The train process of CLUB is different from MINE, we only need to sample data from the joint distribution $\mathbb{P}_{\mathcal{SU}}$. Then we directly minimize the loss $\mathcal{L}(\omega_2)$ based on the samples to improve the accuracy of upper bound of MI estimated by CLUB.

## D  INTEGRATE PMIC WITH RODE

RODE is a role based algorithm which decomposes joint action spaces into restricted role action space to reduce the primitive action-observation spaces. In RODE, each task can be decomposed into a sub-task which has a smaller action-observation space, each sub-task is associated with a role $\rho$. To demonstrate the generalization and effectiveness of PMIC, we design PMIC-RODE (e.q., integrate PMIC with RODE). To integrate with PMIC, we take the joint role $\boldsymbol{\rho}$ into consideration. Thus we

measure the MI between the joint policy and global state with the joint role $\boldsymbol{\rho}$ (e.q., $I(u; \boldsymbol{\rho}, s)$). Since the action is selected based on the observation and the selected role. $I(u; \boldsymbol{\rho}, s)$ means the correlation between the global state, the selected joint role and joint policy which has the same effort with $I(u; s)$ in PMIC-MADDPG. Different from PMIC-MADDPG, we set $r_t^{\mathrm{DMIE}} = \alpha I_{\mathrm{MINE}}(s_t, \boldsymbol{\rho}_t; u_t) - \beta I_{\mathrm{CLUB}}(s_t, \boldsymbol{\rho}_t; u_t))$. $r_t^{\mathrm{DMIE}}$ is used in the same way as in PMIC-MADDPG. To achieve the goal in Equation 5, we only need to modify the $Q_{tot}$ as follows:

$$\mathcal{L} = \mathbb{E}_{\mathcal{D}} \left[ \left( y_{rode} - Q_{tot}(s, \boldsymbol{u}) \right)^2 \right], \tag{7}$$

where $y_{rode} = r + r^{\mathrm{DMIE}} + \gamma \max_{a'} \bar{Q}_{tot} \left( s', \pi_{\theta'}(s') \right)$ and $Q_{tot}$ is a QMIX-style (Rashid et al., 2018) mixing network to estimate the global action-value and $\bar{Q}_{tot}$ is the target network of $Q_{tot}$. DMIE is updated according to Equation 4. Since RODE is QMIX-based method, credit assignment is also applied to $I_{\mathrm{MINE}}$ and $I_{\mathrm{CLUB}}$. The MI on superior collaboration patterns and inferior collaboration patterns can reward and punish the agents based on their contribution, which can better help to find the optimal collaboration patterns.

To integrate PMIC with other algorithms, we simply replace MADDPG with other MARL algorithms. For example, To integrate with RODE, we only need to initialize the networks of RODE in line 1 of Algorithm 1 and retain other components. Then we use RODE to select actions in line 4-5. All other processes remain the same except changing $I(u; s)$ to $I(u; \boldsymbol{\rho}, s)$. Finally, we update the value network according to Equation 7 and others are the same with RODE to replace the line 12 in Algorithm 1. The same applies in combination with other algorithms.

## E  EXPERIMENTAL SETTINGS

On MPE and SMAC, we use episode reward as the criterion for adding data to DPCB.

On Multi-Agent MuJoCo, we can more accurately collect superior collaboration patterns. Since when the episode reward of one trajectory is high, there is no guarantee that there are no poor sub-trajectories which affect DPCB. Thus we leverage a more fine-grained filtering method to filter the collaboration patterns on Multi-Agent MuJoCo. We leverage sub-trajectories to replace the complete trajectory to accurately filter superior and inferior collaboration patterns. In particular, we set the length of sub-trajectories to 50 steps except 100 steps for 2-Agent Walker and use the total reward of each sub-trajectory as the criterion for adding data to DPCB. The update frequency of MINE and CLUB is 1 on all tasks.

The experiments on MPE are carried out on Intel(R) Xeon(R) Gold 5117 CPU @ 2.00GHz. The experiments on Multi-Agent MuJoCo are carried out on NVIDIA GTX 2080 Ti GPU with Intel(R) Xeon(R) CPU E5-2680 v4 @ 2.40GHz. The experiments on SMAC are carried out on Intel(R) Xeon(R) CPU E5-2680 v4 @ 2.40GHz and Intel(R) Xeon(R) CPU E5-2679 v4 @ 2.50GHz.

## F  ADDITIONAL EXPERIMENTS

In this section, we present more experiments to help better understand our method. Four questions are raised and more experiments on Multi-Agent MuJoCo are provided.

**RQ1**: Is DPCB buffer size sensitive?

**RQ2**: Is maximizing $I(s; \pi(\cdot|s))$ better than other MI forms?

**RQ3**: What is the time consumption of PMIC-MADDPG?

**RQ4**: Is $\alpha$ and $\beta$ of PMIC sensitive?

To answer RQ1, we design experiments on MPE and adjust the size of DPCB. Specifically, we test 5 different buffer sizes (100, 500, 1000, 5000, 10000) for positive buffer and negative buffer of DPCB. The results shown in Figure 13 indicate that both too large buffer size and too small buffer size have a negative effect on the performance. A buffer size that is too small prevents MINE and CLUB from carving patterns well, while too large buffer size hinders the optimality of DPCB which makes it impossible for MINE and CLUB to adapt quickly to the new collaboration patterns. Therefore an appropriate buffer size is important.

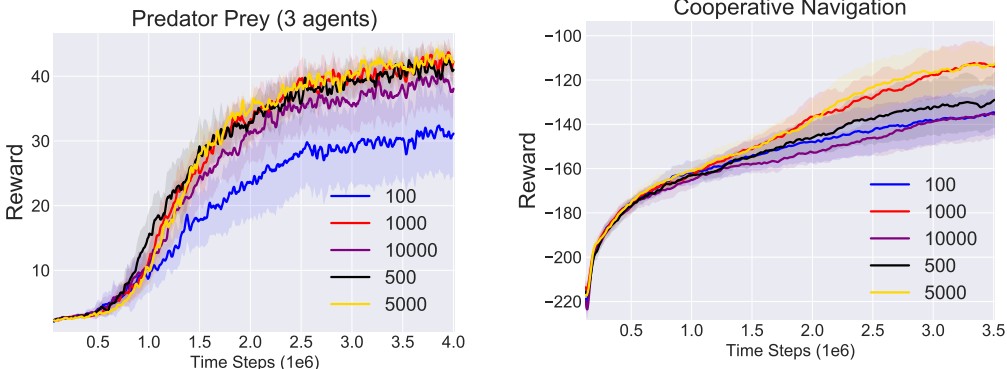

Figure 13: Influence of DPCB size.

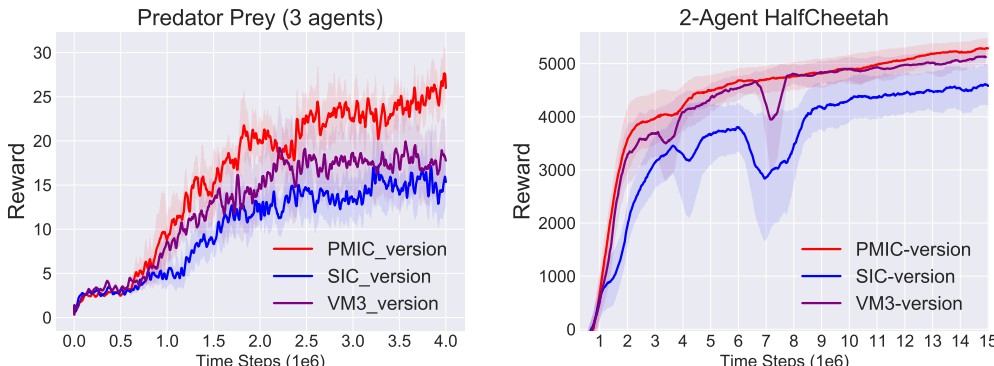

Figure 14: Comparison of maximizing mutual information with different forms. PMIC-version leverages MI of global state and joint action. VM3-version leverages the MI of any two agents' policies among agents. SIC-version leverages the MI of $z$ and the joint policy.

To answer RQ2, we directly maximize MI with different forms to give a comparison. To avoid differences caused by some mechanisms (e.g., double Q) or structures (e.g., MINE), we apply different forms of MI to MADDPG and leverage MINE as the MI estimator and other settings remain the same. The hyperparameters have been fine-tuned. The results are shown in Figure 14. Maximize MI of the global state and the joint action is better than other MI forms in terms of the final performance. This demonstrates that our proposed method of maximizing the MI of global state and joint action is more effective. Besides, we also find that maximizing MI on environments with sparse reward brings performance degradation to the original algorithm. The reason is: Due to the difficulty of exploration on environments with sparse reward, mutual information will play a greater role in guiding compared to sparse reward. Since maximizing MI has the problem of easily making agents fall into suboptimal collaboration, agents quickly fall into suboptimal collaboration with the guidance of MI.

To answer RQ3, we evaluate the time consumption of different algorithms on 6-Agent HalfCheetah. The experiment is carried out on NVIDIA GTX 2080 Ti GPU with Intel(R) Xeon(R) CPU E5-2680 v4 @ 2.40GHz. We evaluate the time consumption of each algorithm individually with no additional programs running on the device. Each result is the average of ten time-consuming calculations. The time consumption includes the time consumption of the execution phase and the time consumption of the centralized training phase. The results are shown in Table 2. PMIC-MADDPG brings less time consumption than other methods.

To answer RQ4, we give ablation experiments of $\alpha$ and $\beta$. The results are shown in Figure 15. The experiment proves that appropriately sized values of $\alpha$ and $\beta$ are critical for algorithm improvement.

Table 2: Time consumption of different algorithm on 6-Agent HalfCheetah every 1000 time steps.

| algorithm | PMIC-MADDPG | MADDPG | SIC-MADDPG | VM3-AC |
|---|---|---|---|---|
| seconds | 30.96 | 24.43 | 31.01 | 182.58 |
| algorithm | COMIX | Fac-MADDPG | MASAC | |
| seconds | 34.43 | 87.13 | 41.68 | |

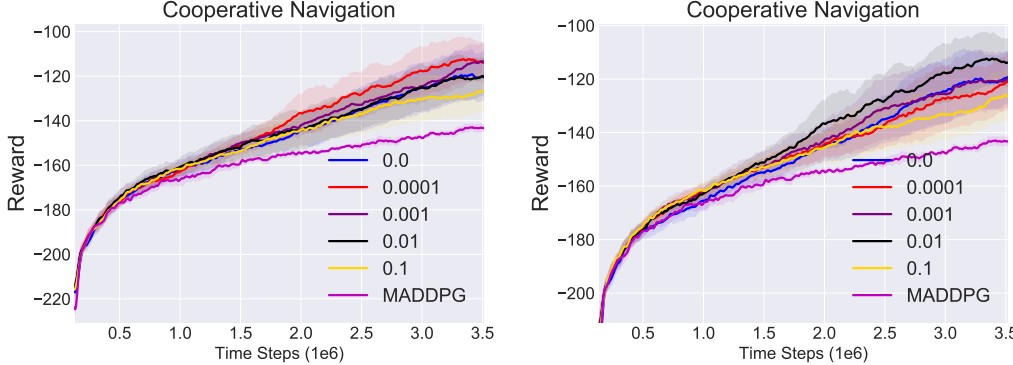

Figure 15: Ablation experiment of $\alpha$ and $\beta$.

The performance loss occurs if $\beta$ or $\alpha$ is too large or too small. Thus for different environments, we need to adjust $\alpha$ and $\beta$ to achieve the best performances.

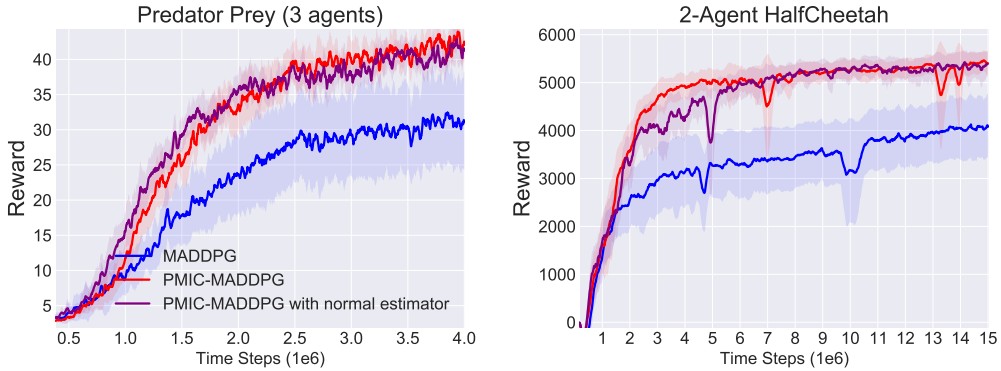

Figure 16: Comparison of PMIC with MINE and PMIC with normal estimator.

Later we evaluate the impact of MINE on the performance of the algorithm. Because we are using MINE for the first time in the field of multi-agent reinforcement learning, thus we need to provide a comparison of MINE and the normal mutual information estimation method. The results are shown in Figure 16. We can see that the performance of both methods is similar, which demonstrates that the effectiveness of PMIC is introduced by the maximization-minimization MI, rather than MINE.

We further analyze whether MINE trained with the positive buffer in DPCB could act as a good guide, which requires the MINE to estimate a large value to superior collaboration patterns and small value to inferior collaboration patterns. We train MADDPG in 2-Agent Walker. During training, we save the positive buffer every 100000 steps, which ensures that the collaboration patterns saved each time is better than the previous time. Then we save the MINE every 100000 steps and use it to estimate MI of collaboration patterns in saved positive buffer. The estimated MI of different positive buffer by different MINE are plotted on Figure 17. The larger the index, the newer the MINE and buffer. When we use DPCB to train MINE, the color gradually darkens from left to right in each line, indicating that MINE has a higher estimate for the better collaboration patterns. But when train MINE in the same way with VM3-AC (Kim et al., 2020), MINE estimates the same value of MI for

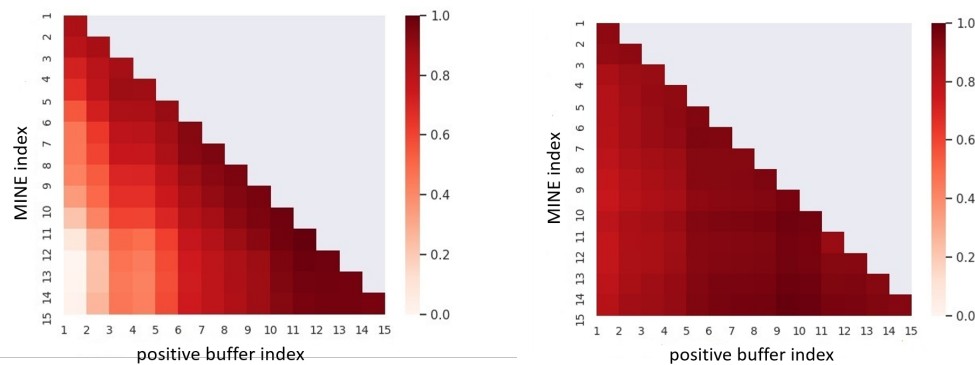

Figure 17: Visualization results of the MI estimated by different MINE for different positive buffer. X-axis denotes the index of positive buffer. Y-axis denotes the index of MINE. The value in coordinates $(x, y)$ represents the MI estimated by the MINE with index y for the buffer with index x.

both inferior and superior collaboration patterns. From the perspective of visualization, the MI of superior coordination estimated by MINE with DPCB is large so that such coordination can be more encouraged, which can give accurate guidance to superior collaboration patterns.

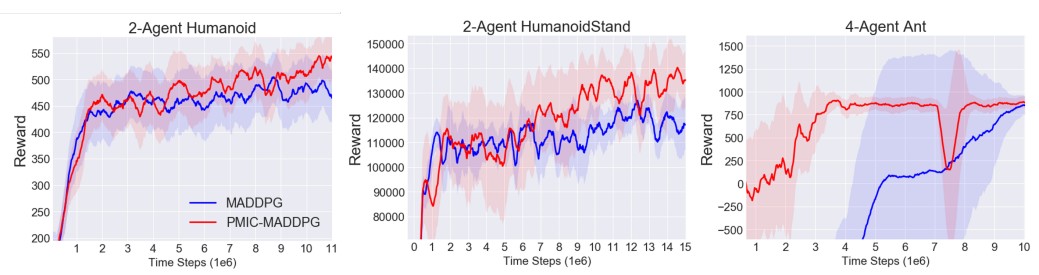

Figure 18: Comparison of more tasks on Multi-Agent MuJoCo.

In addition to the above experiments, we add more experiments on Multi-Agent MuJoCo. The results are shown in Figure 18. PMIC has also significantly improvement on the original algorithm for some more difficult robot control tasks such as humanoid and humanoid-standup. These experiments comprehensively demonstrate the effectiveness and generality of PMIC.

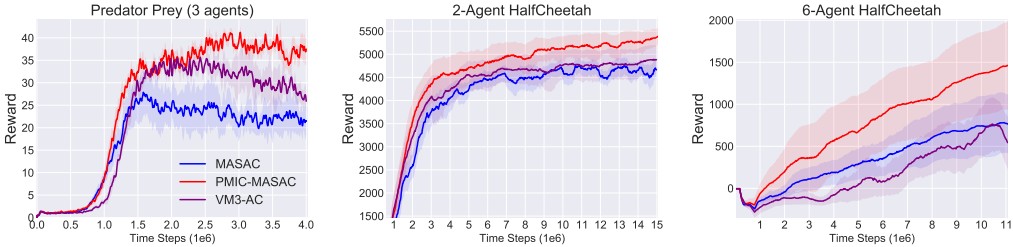

Figure 19: Comparison of MASAC-based methods on Multi-Agent MuJoCo.

Finally, we integrate PMIC with MASAC to further verify the generalisation and effectiveness of PMIC. The results are shown in Figure 19. PMIC has also significantly improvement on MASAC. Beside, PMIC-MASAC is better than other MASAC-based method VM3-AC. The effectiveness and generalisation of PMIC is more convincing by the above experiments.

## G   COMPARISON WITH OTHER RELATED ALGORITHMS

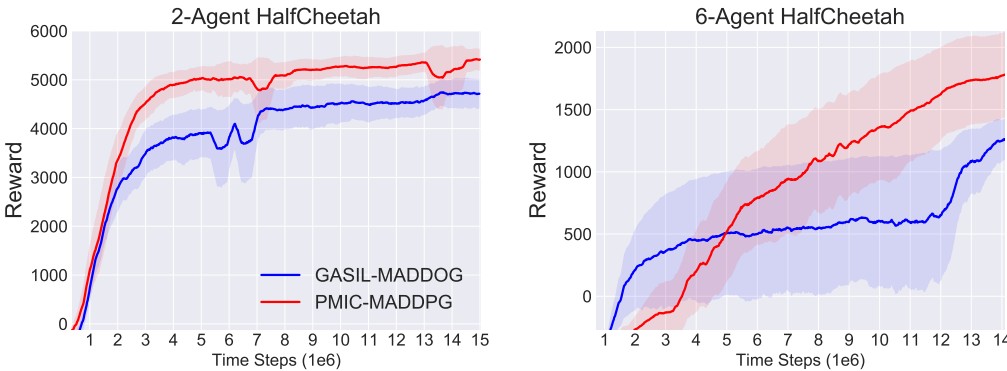

Figure 20: Comparison between leveraging PMIC and GASIL.

In this section, we give a new question: Whether PMIC can be replaced by a discriminator which can also achieve the purpose of guiding agents to better collaboration. To answer the question, more experiments are carried out. This idea using discriminator is similar to Generative Adversarial Self-Imitation Learning (GASIL) (Guo et al., 2018) where a discriminator is trained to discrimiate between superior trajectories and inferior trajectories, while the policy learns to fool the discriminator by imitating superior trajectories. Specifically, GASIL maintains a good trajectory experience. The discriminator scores the good trajectory to 1.0 and scores the trajectory generated by current policies to 0.0. The results are shown in Figure 20 which indicates that our MI based architecture PMIC is more effective than GASIL, which may be mainly due to the fact that the discriminator is difficult to train and can suffer from pattern collapse.

## H   HYPERPARAMETERS

The hyperparameters for network architecture:

1: On MPE, we use two fully connected layers comprised of units 64 with ReLU nonlinearity and a final layer with tanh to output actions as the policy network for each agent, use the critic network with the same architecture as the policy network except tanh for the final layer.

2: For Multi-Agent MuJoCo, the policy networks and critic network use the same architecture with those on MPE, but with 200 and 100 units for two fully connected layers.

3: For SMAC, we use code provided by RODE, the parameters and details are the same with the original paper.

All algorithms' network architecture (on MPE and Multi-Agent MuJoCo) is the same to ensure that the comparison is fair. For MADDPG, we use a centralized Q network and $N$ policy networks. For MINE, we use two fully connected layers (100 units on Muti-Agent MuJoCo and 64 on MPE) with Leaky ReLU nonlinearity to encode global states and joint actions respectively, the results are obtained by dot product of the embeddings of global state and joint action. For CLUB, we use two fully connected layers (50 units on Muti-Agent MuJoCo and 32 on MPE) with Leaky ReLU nonlinearity to encode global states. For SMAC, we need to encode $\rho$, thus CLUB uses two fully connected layers (64 units) to encode $\rho$ and the global state, then the outputs are concatenated to form a vector of 128 dimensions, and use the vector to predict the mean and variance of the global action. MINE uses two fully connected layers (32 units) to encode $\rho$ and the global state and uses a fully connected layer (64 units) to encode the joint action. the results are obtained by dot product of the embeddings of new vector (e.q., combine the vectors of the global state and the joint role $\rho$) and joint action). The architectures used to calculate MI in other MI-related algorithms (SIC-MADDPG and VM3-AC) use the same number of units and activation function with MINE and the other parts are consistent with the setting in original papers.

For SIC-MADDPG and VM3-AC, there are two parameters to adjust: the dimension of $z$ and $\alpha$. For SIC-MADDPG, we select $z$'s dimension from $[2, 3, 5, 8, 10, 15, 20]$ following the setting in original paper on MPE, $[2, 3, 4, 5, 8, 10, 15]$ on Multi-Agent MuJoCo and select $\alpha$ from $[0.1, 0.01, 0.001, 0.0001, 0.00001]$ following the setting in original paper. For VM3-AC, we select $z$'s dimension from $[2, 4, 8]$ following the setting in original paper and select $\alpha$ from $[0.1, 0.01, 0.001, 0.0001, 0.00001]$. For MASAC, we adjust $\beta$ from $[1.0, 0.1, 0.01, 0.001, 0.0001], 0.00001$ to control the entropy. For PMIC, we need to adjust $\alpha$ and $\beta$, we select from $[1.0, 0.1, 0.01, 0.001, 0.0001]$. The final choice of $\alpha$, $\beta$ and dimension of $z$ is shown in Table 3. For FacMADDPG and COMIX, we use the official code and the parameters of the original paper.

For other hyperparameters on MPE and Multi-Agent MuJoCo, $1 \times 10^{-3}$ for critic and $1 \times 10^{-4}$ for actor on MPE except $1 \times 10^{-2}$ on Wildlife Rescue and use Adam optimizer with learning rate $1 \times 10^{-3}$ for critic and $1 \times 10^{-4}$ for actor on Multi-Agent MuJoCo. For MINE and CLUB, the learning rate is $1 \times 10^{-4}$ on all environments except $1 \times 10^{-3}$ on Wildlife Rescue. The discounted factor $\gamma$ and $\tau$ are 0.99 and 0.002 on Multi-Agent MuJoCo and 0.95 and 0.001 on MPE. Replay buffer size is $1 \times 10^{6}$ on MPE and Multi-Agent MuJoCo excpet $3 \times 10^{5}$ on Wildlife Rescue. Batch size is 1024 on MPE and 100 on Multi-Agent MuJoCo. The size of positive buffer and negative buffer of DPCB is 1000 on MPE except 6000 on Wildlife Rescue. 5000 for positive buffer and negative buffer on Multi-Agent MuJoCo. 500 for positive buffer and 3000 for negative buffer on SMAC.

For hyperparameters of RODE, all parameters remain the same as in the code provided in the original paper. RODE has two main adjusted hyperparameters: Number of role clusters and role interval. Number of role clusters is used to control the number of role types. The role interval decides how frequently the action spaces change and may have a critical influence on the performance. All parameter settings are consistent with the original paper. The selection of $\alpha$ and $\beta$ is shown in Table. 4. Batch size is 128 to update MINE and CLUB. We apply maximization and minimization of MI after 100000 time steps on SMAC.

Table 3: Selection of $\alpha$, $\beta$ and dimension of $z$ for different algorithms on MPE and Multi-Agent MuJoCo.

|  | SIC-MADDPG | PMIC-MADDPG | VM3-AC | MASAC |
|---|---|---|---|---|
| Env name | $\alpha\|z$ dim | $\alpha\|\beta$ | $\alpha\|z$ dim | $\beta$ |
| Predator Prey (3 Agents) | 0.00001 \| 2 | 0.01 \| 0.1 | 0.1 \| 4 | 0.1 |
| Predator Prey (6 Agents) | 0.0001 \| 8 | 0.01 \| 0.1 | 0.01 \| 8 | 0.1 |
| Predator Prey (12 Agents) | 0.0001 \| 3 | 0.1 \| 0.1 | 0.01 \| 4 | 0.1 |
| Predator Prey (24 Agents) | 0.0001 \| 2 | 0.01 \| 0.1 | 0.01 \| 4 | 0.1 |
| Cooperative Navigation | 0.0001 \| 2 | 0.0001 \| 0.01 | 0.1 \| 4 | 0.01 |
| Wildlife Rescue | 0.001 \| 3 | 0.001 \| 0.1 | 0.001 \| 2 | 0.001 |
| 2 agents HalfCheetah | 0.0001 \| 3 | 0.1 \| 0.0001 | 0.1 \| 8 | 0.01 |
| 6 agents HalfCheetah | 0.0001 \| 2 | 0.1 \| 0.001 | 0.1 \| 2 | 0.01 |
| 2 agents Walker | 0.0001 \| 3 | 0.1\| 0.001 | 0.1 \| 8 | 0.01 |

Table 4: Selection of $\alpha$, $\beta$ on SMAC.

|  | PMIC-RODE |
|---|---|
| Map name | $\alpha\|\beta$ |
| MMM2 | 0.0001 \| 0.0001 |
| MMM | 0.0001 \| 0.001 |
| 2s3z | 0.001 \| 0.1 |
| 3s_vs_5z | 0.001 \| 0.01 |
| 3s5z | 0.001 \| 0.01 |
| 10_vs_11m | 0.01 \| 0.01 |

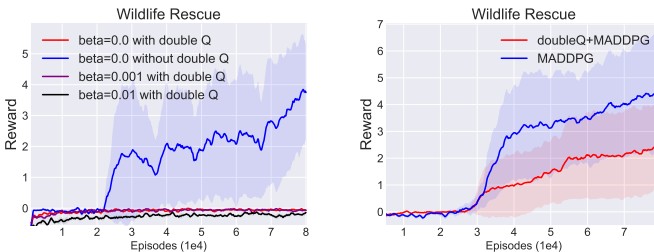

Figure 21: Ablation experiments of $\beta H$ and double Q on Wildlife Rescue.

# I   EXPERIMENTS ABOUT MASAC-RELATED ALGORITHMS ON WILDLIFE RESCUE

Through experiments, we find that the MASAC-related algorithms can not learn positive rewards on Wildlife Rescue environment. Thus we experiment MASAC on Wildlife Rescue environment to find the reason.

There are two differences between MASAC and MADDPG: $\beta H(\pi)$ and double Q mechanism. Firstly, we make adjustments to $\beta$ and find that no matter how we adjust $\beta$, we can not get positive rewards, even if $\beta$ is set to 0.0. Secondly, we change the network architecture of MASAC by removing the double Q mechanism, then find that the algorithm can get positive rewards. Thus we hypothesize that the main reason why MASAC-related algorithms can not learn to cooperate on Wildlife Rescue is caused by double Q mechanism.

To further verify our hypothesis, we add double Q to MADDPG and find significant performance degradation and slower convergence which proves that double Q mechanism is the main factor. We give some explanations, the Wildlife Rescue is characterized by high punishment for miss-coordination and low positive rewards. Double Q prevents overestimation but can lead to underestimation, thus using double Q might ignores less frequent positive rewards, leading to underestimation.

