# OpenReview forum: "PMIC: Improving Multi-Agent Reinforcement Learning with Progressive Mutual Information Collaboration"
_ICLR.cc/2022/Conference — ICLR 2022 Submitted_

### Official Review · Reviewer_iVbL · 2021-10-30

**Correctness:** 3
**Technical Novelty And Significance:** 3
**Empirical Novelty And Significance:** 2
**Recommendation:** 6
**Confidence:** 4

**Details Of Ethics Concerns:**

No.

**Main Review:**

## Strengths

1. The writing of this paper is generally clear.
2. The motivation of this paper is clear that aims to solves the problem that is observed from the existing state-of-the-art algorithms.
3. The experiments are sufficient, including the discrete/continuous action cases, different environments/tasks and multiple ablation studies to verify the effectiveness of the proposed framework.
4. The proposed framework can be incorporated into any MARL algorithms, so it is a general method.

## Weaknesses

This paper does not have obvious weaknesses, but I have some concerns that the authors need to clarify.
1. Can the authors give more details about how $\mathbb{P}\_{ \mathcal{S} \mathcal{U} }$, $\mathbb{P}\_{\mathcal{U}}$ and $\mathbb{P}\_{\mathcal{S}}$ implemented?
2. In experiments, the authors claim that the results from the first two graphs in Figure 8 shows that the proposed maximizing and minimizing MI can improve the stability. However, the stability of the proposed method is not well, i.e., the variance of the proposed method is much larger than other competitors. Can the authors give more explanations on this?
3. For the abalation study in Figure 9, the authors did not show the case that PMIC-MADDPG-normal\_buffer that can directly show the effectiveness of the proposed replay buffer. Can the authors show the result for it?
4. If I understand correctly, $T\_{\omega\_{1}}$ is designed with the state and action encoders while $T\_{\omega\_{2}}$ is without. Can the authors explain the reason for the inconsistent design for these two models? It is better that the authors can provide the results for MINE without state/action encoders to get rid of the suspect that the performance improvement relies on the encoders rather than the main contribution claimed in the paper.

## Minors

There are multiple writing typos in the paper.

1. "Similarly, CLUB approximates the lower bound ..." where lower bound should be upper bound.
2. $I(s; \pi(u|s)$ appearing in multiple places should be $I(s; \pi(u|s) )$.
3. Above Equation (3), $D\_{a}(\cdot)$ should be $D\_{u}(\cdot)$.

The above are just the ones I found. I urge the authors to check the paper writing again, and revise these writing typos in the discussion stage.

**Summary Of The Paper:**

This paper aims to solve the collaboration among agents (that is usually called forced coordination in other literatures). The common method to enhance the correlations between agents by mutual information (MI) could lead to the sub-optimal collaboration. To address this issue, the author propose Progressive Mutual Information Collaboration (PMIC). This new MARL framework is composed of two components: (1) two replay buffers that respectively collect superior trajectories and inferior trajectories; (2) two mutual information estimators that respectively estimate the upper bound and the lower bound of MI that are trained with the data stored in the two replay buffers introduced above. These two estimators (MINE and CLUB) are from the prior works. The novelties of this work are applying these two estimators to collaboration problem for MARL and learning the mutual information that evaluates the correlation among agents constrained within these two bounds. In my view, the idea of appropriately modelling these MI bound estimators to the actual phenomenon in the MARL problem is novel. Seen from the algorithmic framework type, this algorithm add the learned MI term to the reward function that plays the role of auxiliary reward. From the perspective of optimization, this is a kind of implementation of Lagrangian method, to transform the constraints stated in Equation (2) and (3) to a unconstrained terms involved in the objective function. Similar to many prior works, the tuning of the multipliers could be a potential issue. The authors also conduct the ablation studies on these two multipliers and the results seem like it would not affect the performance in the case of Cooperation Navigation.

**Summary Of The Review:**

In summary, this paper has some novolty but stays at the application level, though this is not my concern. My main concern is the effectiveness of the main contribution claimed in the paper, i.e. the maximization-minimization MI. I concern that whether the performance improvement mainly comes from the specific architecture $T$ used in the framework.

For the current version of paper, I can only recommend reject (but marginally below the threshold) due to the weakneses and concerns above.

---

> ### Author Response · Authors · 2021-11-18
> **Initial Response to Reviewer iVbL**
>
> We sincerely appreciate your responsible reviews and your constructive comments, and we would like you to know that your questions provide considerably helpful guidance to improve the quality of our paper.
>
> We will address each of the issues raised by the reviewer below:
>
> (1) [Re: “details about how $\mathbb{P}\_{\mathcal{S}{\mathcal{U}}}$ and $\mathbb{P}\_{\mathcal{S}}$ $\mathbb{P}\_{\mathcal{U}}$  implemented?”]
>
> $\mathbb{P}\_{\mathcal{S}{\mathcal{U}}}$ is state-action joint distribution which is composed of $s$-$u$ pairs in the positive buffer or negative buffer of DPCB. $\mathbb{P}\_{\mathcal{S}}$ and $\mathbb{P}\_{\mathcal{U}}$ are the marginal distributions of $\mathbb{P}\_{\mathcal{S}{\mathcal{U}}}$ which are composed of the $s$ and $u$ separately in the positive buffer or negative buffer of DPCB.
>
> (2) [Re: "the variance of the proposed method is much larger than other competitors. Can the authors give more explanations on this?"]
>
> The variances of the algorithms on Wildlife Rescue environment are larger than the variances in other environments, which may be caused by the fact that the environment requires higher degree of collaboration and is more likely to make algorithms being stuck into sub-optimal. Other algorithms also have large variances in this environment shown in Figure 6. Overall, the variances of our algorithm are small in most environments.
> The word “stability” here denotes that in most environments of our experiments, our algorithm shows more effective and efficient results than the baseline algorithms. Thanks to the reviewers' comments, we have rephrased this to make the expression more accurate.
>
>
> (3) [Re: "the authors did not show the case that PMIC-MADDPG-normal-buffer that can directly show the effectiveness of the proposed replay buffer."]
>
> According to the comment, we have added the performance of PMIC-MADDPG with normal buffer in Figure 10. This experimental result further demonstrates the importance of discriminativly maximizing and minimizing mutual information, and also demonstrates the effectiveness of DPCB.
>
>
> (4) [Re: "concern that whether the performance improvement mainly comes from the specific architecture $T$ used in the framework."]
>
> Firstly, we need to emphasise that MINE and CLUB are designed in the same way as in the original paper[1][2] and we have not made any changes. Besides, the structure of MINE in our paper is also the same with the ones in EMI[3] and MUSIC[4]. MINE only serves as a technique for mutual information estimation.
> To exclude the effect of MINE, we replace MINE with the same method of estimating the lower bound of mutual information in SIC and VM3, and the experimental results are shown in Figure 16. Both methods with MINE and with normal method achieve similar performance and all outperform other baselines. This experimental result shows that the performance of PMIC is brought by maximization-minimization MI, not the choice of MINE itself. On the other hand, to fully demonstrate the effectiveness of PMIC, we add experiments which combine PMIC with MASAC, and the results are shown in Figure 19, which further demonstrates the effectiveness and generality of PMIC.
>
> We hope our replies have resolved all the issues you posed and showed the improved quality of the paper. **We are always willing to answer any of your concerns** about our work and we sincerely wish you to value the technical innovation and overall contributions of the paper.
>
> [1] Mohamed Ishmael Belghazi, Aristide Baratin, Sai Rajeshwar, Sherjil Ozair, Yoshua Bengio, Aaron Courville, and Devon Hjelm. Mutual information neural estimation. In International Conference on Machine Learning, pp. 531–540, 2018
>
> [2] Pengyu Cheng, Weituo Hao, Shuyang Dai, Jiachang Liu, Zhe Gan, and Lawrence Carin. Club: A contrastive log-ratio upper bound of mutual information. In International Conference on Machine Learning, pp. 1779–1788. PMLR, 2020.
>
> [3] Hyoungseok Kim, Jaekyeom Kim, Yeonwoo Jeong, Sergey Levine, Hyun Oh Song:
> EMI: Exploration with Mutual Information. ICML 2019: 3360-3369
>
> [4] Rui Zhao, Yang Gao, Pieter Abbeel, Volker Tresp, Wei Xu: Mutual Information State Intrinsic Control. ICLR 2021

---

> > ### Comment · Reviewer_iVbL · 2021-11-21
> > **Re: Initial Response to Reviewer iVbL**
> >
> > The authors have addressed the most of my concerns, i.e., (2), (3) and (4). However, I still have concerns about (1).
> >
> > In (1), the authors may not get my thought. I mean whether $\mathbb{P}\_{\mathcal{SU}}$, $\mathbb{P}\_{\mathcal{S}}$ and $\mathbb{P}\_{\mathcal{U}}$ are implemented in 3 networks?

---

> > > ### Author Response · Authors · 2021-11-21
> > > **Response to POST REBUTTAL 1 Comments from Reviewer iVbL**
> > >
> > >
> > > We appreciate the reviewer's quick response very much!
> > > We explain more about question (1) below:
> > >
> > > We do not explicitly model the three distributions with three networks. In practice, we approximate sampling from the three distributions by sampling data from the positive/negative buffer in different ways.
> > > Taking MINE for a demonstration here, we randomly sample a batch of $s$-$u$ pairs from the positive buffer to represent the joint distribution $\mathbb{P}\_{\mathcal{S}{\mathcal{U}}}$. Then we sample a batch of $s$ and $u$ separately from the positive buffer and combine them to form $s$-$u$ pairs to represent the joint of marginal distrutions $\mathbb{P}\_{\mathcal{S}}\otimes \mathbb{P}\_{\mathcal{U}}$.
> > > For the case of CLUB, it is the same.
> > > Finally, we leverage these $s$-$u$ pairs to train MINE and CLUB by Equation (2) and (3).
> > > Our implementations are consistent with the original papers' implementations[1][2].
> > >
> > >
> > > We hope our reply have resolved the concern left and
> > > careful amendment on this point and other valuable suggestions will be seriously made and considered in our later revision.
> > >
> > > [1] Hjelm, R. D., Fedorov, A., Lavoie-Marchildon, S., Grewal, K., Trischler, A., and Bengio, Y. Learning deep
> > > representations by mutual information estimation and maximization. arXiv preprint arXiv:1808.06670, 2018.
> > >
> > > [2] Pengyu Cheng, Weituo Hao, Shuyang Dai, Jiachang Liu, Zhe Gan, and Lawrence Carin.  Club:A contrastive log-ratio upper bound of mutual information. In International Conference on MachineLearning, pp. 1779–1788. PMLR, 2020.

---

> > > > ### Comment · Reviewer_iVbL · 2021-11-21
> > > > **Re: Response to POST REBUTTAL 1 Comments from Reviewer iVbL**
> > > >
> > > > Thanks for the author's response. I understand now.

---

### Official Review · Reviewer_5fQS · 2021-11-02

**Correctness:** 3
**Technical Novelty And Significance:** 3
**Empirical Novelty And Significance:** 3
**Recommendation:** 6
**Confidence:** 3

**Main Review:**

Strengths:
The proposed method is novel and shows good performance.

Weaknesses:
1. While the results generally look good, some reward curves in Figure 5, 6, 7, 8, and 9 are not converged. It would be better to compare the converged performance between baselines.
2. I am not quite following why minimizing the MI in inferior trajectories is helping improve the corporation. Generally, I would suggest more explanations on the motivation of the PMIC.

**Summary Of The Paper:**

This paper proposes PMIC a MARL framework for improving multi-agent collaboration through mutual information. PMIC uses two separate buffers to store the superior and inferior trajectories, and uses them to train two estimators for the upper and lower bounds for the mutual information between global state and joint policy in two sets of trajectories. The estimated mutual information bounds are then used as additional rewards for training individual policies. The experiments presented in the paper show the advantage over several baselines on several benchmark environments.

**Summary Of The Review:**

I think the submission overall is good for its novelty and experiment results. I only have some minor concerns about the theoretical backup of the proposed method. Thus, I suggest accepting the paper.

---

> ### Author Response · Authors · 2021-11-18
> **Initial Response to Reviewer 5fQS**
>
> We appreciate the reviewer's suggestion, we are running the associated experiments for longer time to provide the converged results. In addition, to make the experiments more convincing, we have added additional experiments to further demonstrate the effectiveness and generality of PMIC. Specifically, we integrate PMIC with MASAC and evaluate on several environments. The results are shown in Figure 19. By comparison, PMIC can deliver significant performance improvements for each of these baseline methods.
>
> For the second concern of the reviewer, minimizing mutual information in inferior trajectories does not directly facilitate agents collaboration. On the contrary, minimizing MI punishes the agents associated with the inferior collaboration patterns and increases the probability of exploring superior collaboration patterns which will be added to the positive buffer of DPCB for further use in policy guidance. In general, minimizing MI promotes agents exploration and makes agents avoid being stuck into inferior patterns.
> Eventually agents can continue to break inferior collaboration patterns and learn towards the superior ones, creating a progressive MI manner for collaboration. Overall, PMIC creates a progressive mechanism for exploring (i.e., minimizing MI) and exploiting (i.e., maximizing MI) better collaboration patterns.
>
> **If reviewers have any other questions or concerns, please let us know and we will do our best to resolve them**. We hopefully work with the reviewer to improve the overall quality of the paper.

---

### Official Review · Reviewer_QvKR · 2021-11-02

**Correctness:** 3
**Technical Novelty And Significance:** 3
**Empirical Novelty And Significance:** 3
**Recommendation:** 5
**Confidence:** 4

**Main Review:**

Strengths

- This paper addresses an important issue of the MI-based MARL framework where the joint policy converges to sub-optimal.
- The proposed idea, which maximizes MI based on superior samples and minimizes MI based on inferior samples to avoid the joint policy falling into the sub-optimal, is novel.
- The authors provide various ablation studies for better understanding.


Weakness and Questions

-There is a lack of explanation for the collaboration criterion (section 3.2), e.g, as I understand, $s$ is a state variable whose distribution is conditioned on $\pi$.
-For the interpretation of the proposed MI form, the authors decompose the proposed MI into the entropy of join policy ($H(\pi(\cdot|s))$, the negative entropy of an agent's policy ($-H(\pi_i|s)$) and conditional negative entropy of the agent's policy given other agents' policies ($-H(\pi_{-i}|, \pi_i, s)$). The authors claim that maximizing $-H(\pi_{-i}|, \pi_i, s)$ minimizes uncertainty on other agents' policies. However, $H(\pi_{-i}|, \pi_i, s)$ reduces to $H(\pi_{-i}|s)$ since $\pi_{-i}$ and $\pi_i$ are independent given the state. Thus, it seems that the interpretation of the proposed MI form, $I(s;\pi(\cdot|s))$, is not proper. In this regard, can we say that the proposed MI form causes the collaboration?
-The authors minimize $I(s;\pi(\cdot|s))$ for experiences which have low returns, so it encourages the joint action to explore for inferior experiences. This is because the joint action should be uniform to minimize $I(s;\pi(\cdot|s))$. On the other hand, maximizing $I(s;\pi(\cdot|s))$ for experiences that have high returns encourages the joint action to exploit for superior experiences. Thus, in my opinion, the effectiveness of the proposed method comes from the further exploration and exploitation strategy rather than the collaboration strategy. To see where the performance gain comes from, it would be great if the authors compare the proposed method with MA-SAC+DPCB and VM3-AC+DPCB.

**Summary Of The Paper:**

This paper proposes to maximize the MI between the global state and joint policy for promoting collaboration. To achieve better collaboration, a Dual Progressive Collaboration Buffer (DPCB) which stores superior and inferior samples separately is introduced, and then the proposed method maximizes the lower bound on MI using the samples in superior buffer and minimizes the upper bound on MI using the samples in inferior samples.

**Summary Of The Review:**

-The proposed idea is novel, but the author should provide further explanation on collaboration criterion and further experiments to see where the performance gain comes from.

---

> ### Author Response · Authors · 2021-11-18
> **Initial Response to Reviewer QvKR**
>
>
> We sincerely appreciate your responsible reviews, and we would like you to know that your questions provide considerably helpful guidance to improve the quality of our paper.
>
> If we may briefly summarize the discussions that have taken place so far as follows:
>
> "Further explanation on collaboration criterion"
>
> We consider that in ideal case, collaborative policies are not independent of each other although the conventional policy representation adopted in CTDE makes such independence assumption (or limitation).
> This is the reason why VM3-AC and SIC introduce additional latent variable to model such dependence explicitly.
> In this sense, for the collaboration criterion proposed in our paper, we model such dependence in an implicit way, i.e., no additional explicit variable is introduced.
> Therefore, $H(\pi_{-i}\mid\pi_{i},s)$ can not reduce to  $H(\pi_{-i}\mid s)$ in our modeling.
>
> "in my opinion, the effectiveness of the proposed method comes from the further exploration and exploitation strategy rather than the collaboration strategy."
>
> We agree with the reviewer's opinion and we consider the opinion is consistent to our main idea, i.e., progressiveness.
> PMIC relies on a progressive mechanism for exploring better collaboration patterns and exploiting superior collaboration patterns.
> This is done respectively in PMIC by minimizing MI to break inferior collaboration patterns
> and maximizing MI to encourage superior collaboration patterns.
>
>
> “Further experiments on MASAC+DPCB and VM3-AC+DPCB to see where the performance gain comes from.”
>
> Does it mean MASAC+PMIC and VM3-AC+PMIC? MASAC and VM3-AC can not be integrated with DPCB directly, because DPCB is one component of PMIC, which can not be used in isolation. Moreover, it may be strange to combine VM3-AC and PMIC since the principles and methods of them are not orthogonal.
>
> Therefore, we provide additional experiments on the combination of PMIC with MASAC. The main experimental environments are Predator Prey and 2-Agent Halfcheetach and 6-Agent Halfcheetach. The experimental results are shown in Figure 19, which mainly demonstrate that the algorithm performance gains are derived from our proposed PMIC, not the difference of baselines.
>
> we hope our replies have resolved all the issues you posed and showed the improved quality of the paper. **We are always willing to answer any of your concerns** about our work and we sincerely wish you to value the technical innovation and overall contributions of the paper.

---

> > ### Comment · Reviewer_QvKR · 2021-11-29
> > **Response**
> >
> > Thank you for the author's responses.
> >
> > 1. VM3-AC+PMIC means that we increase the MI between actions for superior samples and decrease the MI between actions for inferior samples. The reason why I suggested the comparison with VM3-AC is to check if the proposed MI form, I(s, \pi(a|s)), is better than I(a^1;a^2), which is proposed in [1].
> >
> > 2. I cannot still understand that $H(\pi^i|\pi^{-i},s)$ cannot reduce to  $H(\pi^i|s)$. As I understand, the authors assume independent policies. The authors mentioned, "we model such dependence in an implicit way" in the author's response. I am wondering how the authors make the implicit dependence. Just modeling a neural network that takes $\pi^{-i}$ and $s$ as input and outputs $\pi^i$ cannot make implicit independence.
> >
> > [1] Woojun Kim, Whiyoung Jung, Myungsik Cho, and Youngchul Sung. A maximum mutual information framework for multi-agent reinforcement learning. arXiv preprint arXiv:2006.02732, 2020

---

> > > ### Author Response · Authors · 2021-11-29
> > > **Response to POST REBUTTAL 1 Comments from Reviewer QvKR**
> > >
> > > We appreciate the reviewer's valuable response.
> > >
> > >
> > > [Re: "The reason why I suggested the comparison with VM3-AC is to check if the proposed MI form, $I(s, \pi(a|s))$, is better than $I(a^1;a^2)$,"]
> > >
> > > We have experiments that verify that $I(s;\pi(u|s))$ is better than $I(\pi_i(\cdot|s);\pi_j(\cdot|s))$. In the experiments, **everything remains consistent except for MI forms**. We give a discussion and provide results in Figure 14 which demonstrate that $I(s;\pi(u|s))$ is more effective in maximizing MI compared with the MI form $I(\pi_i(\cdot|s);\pi_j(\cdot|s))$ proposed in VM3-AC. In addition, we may highlight that $I(\pi_i(\cdot|s);\pi_j(\cdot|s))$ suffers from **the scalability issue** and need to introduce **additional shared inputs** (as we mentioned in Sec.1 and Sec.3.2). which make $I(\pi_i(\cdot|s);\pi_j(\cdot|s))$ impossible to apply in many scenarios. For example, In the scenario with **6 agents (6-Agent HalfCheetach)**, time consumption of different algorithms is shown in Table 2, VM3-AC with $I(\pi_i(\cdot|s);\pi_j(\cdot|s))$ requires nearly **4 times additional time overhead** compared to other methods. On the contrary, $I(s;\pi(u|s))$ is simple and does not have these limitations. That's why we use $I(s;\pi(u|s))$ as a measure of collaboration instead of $I(\pi_i(\cdot|s);\pi_j(\cdot|s))$.
> > >
> > >
> > >
> > >
> > > **As the discussion phase will end in several hours**, we are afraid that we are not able to provide more additional experimental results for the comparison.
> > > We still appreciate the reviewer's constructive suggestions and we will consider to add this comparison in our later version.
> > >
> > >
> > > [Re: "I cannot still understand that $H(\pi_{-i}|\pi_{i},s)$ cannot reduce to $H(\pi_{-i}|s)$".]
> > >
> > > First, $I(s; \pi(u|s))$ is a qualitative measurement for $s,u$ data generated by any policies.
> > > How each term decomposed from $I(s; \pi(u|s))$ is modeled or represented by specific functions, is for the (high-quality, accurate) qualitative estimate.
> > > And it is not associated with how agents' policies are constructed and represented directly.
> > > Second, note that the estimates of $I(s; \pi(u|s))$ serves as **rewards**.
> > > For cooperative agents typically under the paradigm of CTDE, agents' policies are updated jointly with the centralized critic towards higher joint-policy values and thus
> > > the collaboration implicitly emerges in the policies of agents.
> > > In the framework of PMIC, the estimates of $I(s; \pi(u|s))$ are treated as auxiliary rewards which are involved in the learning of the centralized critic;
> > > and they then guide the policy learning during the centralized training.
> > > Therefore, we consider that such dependence among agents are modeled in an implicit way under the framework of PMIC.

---

> > > > ### Comment · Reviewer_QvKR · 2021-11-30
> > > > **Response**
> > > >
> > > > Thank you for the author's responses.
> > > >
> > > > I now understand the effectiveness of the proposed method compared to VM3-AC.
> > > > However, it is unclear that "I cannot still understand that $H(\pi_{-i}|\pi_i, s)$ cannot reduce to $H(\pi_{-i}|s)$."
> > > >
> > > > I understand $I(s;\pi(u|s))$ is a kind of collaboration measurement.
> > > > The authors decompose $I(s;\pi(u|s))$ into three terms to analyze the proposed MI form.
> > > > $H(\pi_{-i}|\pi_i, s)$ is one of them, but I think $H(\pi_{-i}|\pi_i, s)$ reduces to $H(\pi_{-i}|s)$ since $\pi_i$ and $\pi_{-i}$ are independent. Then, the properties the author mentioned in the paper should be modified.
> > > > It would be great if the authors explain why $H(\pi_{-i}|\pi_i, s)$ cannot reduce to $H(\pi_{-i}|s)$. Do $\pi_i$ and $\pi_{-i}$ are not independent?

---

> > > > > ### Author Response · Authors · 2021-11-30
> > > > > **Response to POST REBUTTAL Comments from Reviewer QvKR**
> > > > >
> > > > >
> > > > > We appreciate the reviewer's quick response very much!
> > > > > For the remaining concern, we provide some more discussion on it below.
> > > > >
> > > > > We have to clarify that, when we discuss $I(s;\pi(u|s))$ in Section 3.2, we consider the general case of multiple agents' policies. We agree that $-H(\pi_{-i}|\pi_{i},s)$ reduces to $-H(\pi_{-i}|s)$ if $\pi_{i}$ and $\pi_{-i}$ are independent. However, in the general case considered by us, $\pi_{i}$ and $\pi_{-i}$ are not necessarily independent,
> > > > > since the actions made by desired collaborative policies are dependent on each other intuitively.
> > > > >
> > > > > We now see that it may be kind of misleading due to the policy representation $\pi(\cdot|s)$ used in Section 3.2. The reader may improperly connect this to the implementation of decentralized policy execution in CTDE we introduce in Section 3.4. As we mentioned in previous response, the general policy forms considered in the introduction of $I(s;\pi(u|s))$ is not associated with how collaborative policies are implemented directly. We will add more explanation for better clarity in our later version.
> > > > >
> > > > >
> > > > > One more thing to note is, in CTDE, agents update their policies together with respect to the centralized critic rather than update independently. We consider that the directions of policy learning provided by the centralized critic can be viewed as signals to encourage collaborative correlation among agents implicitly. This is also the way we make the MI estimates into effect in the CTDE implementation.

---

> ### Author Response · Authors · 2021-11-27
> **To reviewer QvKR: Further explanation on Mothodology and Others**
>
> We tried to re-understand the review carefully and provide some additional explanation and clarification below.
>
> (1) [Re: “The authors minimize $I(s;\pi(u|s))$ for experiences which have low returns, so it encourages the joint action to explore for inferior experiences”]
>
> We need to clarify that, minimizing $I(s;\pi(u|s))$ for inferior experiences is to break inferior collaboration patterns,
> which can be interpreted as encouraging agents to explore non-inferior collaboration or superior collaboration instead of exploring for inferior collaboration. By making such asymmetric MI minimization and maximization into effect, agents are expected to keep breaking inferior collaboration patterns and learning towards the superior ones, progressively improving the collaboration performance.
>
>
> (2) [Re: "then the proposed method maximizes the lower bound on MI using the samples in superior buffer and minimizes the upper bound on MI using the samples in inferior samples."]
>
> To avoid potential misunderstanding, we make some additional clarification on 1) the training of MINE/CLUB and 2) the utilization of MINE/CLUB for progressive collaboration guidance of agents.
>
> For 1), the optimization of MINE and CLUB uses the samples in positive buffer and negative buffer of DPCB separately, which is separate from the optimization of policies. After training MINE/CLUB with DPCB, MINE/CLUB provides an estimate of the lower/upper bound of MI, which can be expressed as a quantitative measure of whether any other new patterns match the joint distribution which is used to train MINE/CLUB (the reason why the bound of MI can be expressed as a quantitative measure is shown in (5) and (6) of the replay to the first reviewer vXVK). Maximizing MI estimates provided by MINE means leading agents towards the distribution of superior patterns; while minimizing MI estimates provided by CLUB means moving away from the distribution of inferior patterns.
>
>
> For 2), we use the MI estimates provided by trained MINE as rewards (to maximize) and use the MI estimates provided by CLUB as penalties (to minimize). Such derived rewards and penalties together work as collaboration guidance of agents, and they are calculated for any experiences in the replay buffer rather than for the positive buffer and negative buffer respectively.
>
>
> (3) [Re: "Further explanation on collaboration criterion"]
>
> In order to form collaboration, we design the collaboration criterion from three perspectives: a holistic perspective, an individual perspective, an inter-agent perspective.
> Specifically, maximizing $H(\pi(\cdot|s))$ encourages agents to explore in the joint action space to enrich the diversity of the collaboration pattern\textcolor{orange}{, which results in multiple collaboration patterns given the global state}.
> Considering practical collaboration problems where there are multiple collaboration patterns given the global state,
> maximizing $-H(\pi_{i}|s)$ may encourage agent's policy to concentrate its probability mass on several collaboration patterns. However, the specific collaboration can not be defined because
> the other agents' behaviors can not be determined.
> Thus, maximizing $-H(\pi_{-i}\mid\pi_{i},s)$ is to model the dependence among agents, i.e., the information on how other agents are likely to behave if one chooses a specific choice of collaboration pattern, which reduces the uncertainty of the agent about the policy of other agents.
> Then the specific coordination can be defined as other agents’ behaviors being informed by the agent $i$ implicitly [1].
> Therefore, we resort to these three information entropy-based objectives above to measure the collaboration degree in our paper.
>
>
>
> (4) For MASAC+DPCB and VM3-AC+DPCB, we're not sure what the reviewer suggests we compare.
> 1) If MASAC+DPCB and VM3-AC+DPCB mean MASAC+PMIC and VM3-AC+PMIC, we have given a reply in the initial response. The experiments in Figure 19 have shown that PMIC brings significant improvement to MASAC and PMIC-MASAC outperforms VM3-AC and MASAC, which demonstrate the effectiveness and generalization of PMIC.
>
> 2) If the reviewer is concerned about whether $I(s;\pi(u|s))$ facilitates collaboration better than other MI forms. We give a discussion and provide experiments in Figure 14 that $I(s;\pi(u|s))$ is more effective in maximizing MI compared with other MI forms for collaboration.
> Beside, maximizing and minimizing $I(s;\pi(u|s))$ and other experiments about $I(s;\pi(u|s))$ have demonstrated the effectiveness of $I(s;\pi(u|s))$ in Figure 9.
>
> We would appreciate it if the reviewer could give us a more clear suggestion and let us know if our response addressed some/all concerns. **We will try our best to address any concerns before the discussion period ends**.
>
>
> [1] Julien Roy, Paul Barde, Felix G Harvey, Derek Nowrouzezahrai, and Christopher Pal. Promoting ´
> coordination through policy regularization in multi-agent deep reinforcement learning. arXiv
> preprint arXiv:1908.02269, 2019.

---

### Official Review · Reviewer_vXVk · 2021-11-02

**Correctness:** 4
**Technical Novelty And Significance:** 3
**Empirical Novelty And Significance:** 3
**Recommendation:** 6
**Confidence:** 4

**Main Review:**

## Post-Rebuttal
I now fully understand the algorithm. Thank the authors for the detailed response. I misunderstood the motivation of using the two bounds. Now the algorithm design looks pretty clear and intuitive to me, which is simply reducing MI on poor trajectories while increasing MI on good trajectories.

**some suggestion on writing**
1. I think the position of Sec. 3.2 makes it a bit misleading since it is presented after the motivating example (Sec. 3.1).  Sec. 3.1 has already assumed the optimizing objective for cooperation is MI and presents the motivating example based on this assumption. Note Sec 3.2 simply re-stated why MI can be a good optimization objective for learning, which is, in fact, the assumption in the previous subsection! Wouldn't it be more appropriate to justify the assumption before assuming it? This cyclic statement would make the presentation less consistent for the readers (including me) to follow. From the best readability, I would suggest the authors have a separate preliminary/motivation section, which contains the current content of Sec 3.2 first (to first show why optimizing MI is good) and then the content of Sec 3.1 (to show the issues of directly optimizing MI when having multiple patterns).

1. I would still encourage the authors to make the best efforts to add the related work section back to the main paper. This paper is related to a lot of good ideas in existing works. It would be critical to mention them directly. The authors may possibly make the experiment sections a bit more concise and shrink the conclusion section to make some space.


=======================================================================

### Strong Points
1. The proposed algorithm is, to the best of my knowledge, novel and it is indeed interesting to see such a separation of buffer with mutual information computation significantly boosts the performance of a collection of MARL algorithms, **with the assumption that the derivation is correct**.
2. There are a lot of ablation studies conducted in the experiment section as well as the appendix, which is appreciated. The experiment section is clear and easy to follow.

### Weak Points
In general, before moving into more detailed comments, I would strongly encourage the authors to substantially re-organize the paper to make it easier for the readers to follow. Also, most of the arguments and motivations in the paper are very hand-wavy, which makes me very concerned about the correctness of the proposed algorithm. It could be because of the writing issue, but, at least from the current statements in the paper, the correctness cannot be rigorously justified. More comments follow.

1. A related work section would be encouraged.

    Sec. 3.1 and the introduction section have a substantial overlap describing the existing literature, which should be put into a separate related work section. Fig.1 also looks strange as it appears in a motivation section. Fig.1 seems to only serve as a role of toy example showing that PMIC works better than some baselines. This is an experiment result rather than a motivation example! It doesn't make sense to me to motivate an algorithm by simply saying hey it works well. As a research paper, a reader would expect insights before moving into empirical evidence of your algorithm being good. I would appreciate it if the authors could rewrite Sec. 3.1 such that some more concrete examples of how the algorithm is working, for example, intuitively what kind of behavior leads to high MI; what characteristics are those sub-optimal strategies intuitively so we have a sense of what kind of policies we want to avoid; and so on. The current Sec.3.1 is simply a replication of the introduction section.

2. I couldn't find how sec. 3.2 and sec 3.3 are related

    > However, simply maximizing this MI can hinder algorithm learning as discussed above.

    Eq.(1) definitely makes a lot of sense to me and there are existing works that consider similar objectives for MARL learning (not exactly the same objective but somewhat similar, like this one for example https://arxiv.org/pdf/2106.02195.pdf ). But why does this objective hinder learning? Is this really discussed? I have made my best efforts on understanding the discussions in Sec.3.1 but I could find any convincing arguments of how this objective hinders learning.

   > Therefore, we propose a novel framework PMIC to solve the problem

   Okay, even if this objective does have hindered learning, then why does the proposed PMIC algorithm solves tackles the issue in this objective? What is the relationship between Eq(1) and PMIC? To be concrete, for example, Sec.3.1 argues that the policy should be diverse (well, the terminology isn't really appropriate but let's use it for now) as suggested by using the entropy bonus but I couldn't find any corresponding term in PMIC for doing so. So what problem does PMIC solve?

   There are so many hand-wavy arguments in this paper like the two above.


3. There are no rigorous justifications on the relationship between high-reward trajectories and MINE/CLUB

    > Intuitively, only the behaviors that obey superior collaboration patterns or inferior collaboration patterns have large MI estimates calculated by MINE or CLUB since $\mathcal{L}(\omega_1)$ and $\mathcal{L}(\omega_2)$ are optimized.

    - *I think this statement is the foundation of the correctness of the method*. If this statement is correct, the whole algorithm becomes neat. Yes, I do have witnessed empirical analysis supporting this argument in Fig 9 and Fig 15. But, I want to emphasize that those two figures are empirical evidence rather than rigorous derivation. Typically, as a research paper, the conventional way of algorithm development is to first provide rigorous deviation (e.g., mathematical proof; equation analysis and approximation; or explanation and discussion on concrete toy examples), then develop an algorithm that solves the formulated problem and finally present empirical supports. The authors simply present this key argument by using the term ''__intuitively__'', which I failed to catch with my best efforts. This argument is the most critical point of the method and if this argument is wrong, the whole paper becomes problematic. Hence, I strongly encourage the authors to provide formal and rigorous justification or proof to this statement.

    - Here is what confuses me a lot. Note that the two buffers are categorized according to their _rewards_ while both MINE and CLUB are bounds for _mutual information (MI)_. So what is the mathematical connection between rewards and MI? Why should superior strategies have higher MINE value? Can you mathematically prove or justify this? To the best of my knowledge, I cannot find any related proof in the existing literature.

    - MINE is a generic lower bound for MI. So wouldn't optimizing MINE with the entire replay buffer strictly lead to a higher lower bound than only using the positive buffer (you have more data!)? Similarly, why not optimize CLUB using the entire buffer as well? Fig 9 does not include such ablation either. Assuming the correctness of all the arguments of MINE and CLUB, using the entire replay buffer should lead to at least the same or probably even better behavior compared with PMIC since you have both tighter lower bound and upper bound?

### Other Issues
1. > the joint policy should be diverse enough to avoid being stuck into sub-optimal collaboration.

    Typically the term "_diverse_" refers to a set of (or a distribution of) different policies rather than a single joint policy. Also, according to the phrase "_avoid being stuck into sub-optimal_", it seems that diverse in fact means that the policy maintains high entropy _throughout the training process_ rather than describing the characteristics of the _final policy_ (the final policy can have low entropy if it converges but during training, it would better have high entropy for the purpose of exploration). So I would suggest the author direct state the purpose of avoiding poor local optimum rather than using the term "_diverse_" (you may be encouraged to refer to the literature of learning diverse skills/strategies in deep/MA RL literature)

2. > 3) Meanwhile, given an agent i’s policy on the global state, the uncertainty on other agents’ policies also should be low, thus more easily to achieve better collaboration

    I don't get why this point is necessary when we have 2) already. I couldn't find any corresponding objective that minimizes the intra-agent policy uncertainty in Sec 3.4. I don't know why this point is presented here.

4. minor issues
    - The predator-prey environment is not a cooperative one, for which the one-sided reward wouldn't become a convincing metric. Are you referring to the cooperative version of predator-prey? I didn't find any content confirming this.

    - I would encourage the authors to have a sub-index for each plot in Fig 8 rather than saying sth like "the third plot in fig.8".

    - A possibly related paper also using mutual information formulation for cooperative MARL: https://arxiv.org/abs/1910.05512

    - Fig 15 is okay but it is not really super convincing. I can still observe some color differences in the right plot (I guess the values range from 0.6 to 1.0 maybe?) So probably you just need some tuning of beta or value normalization to also make VM3-AC work.


**Summary Of The Paper:**

This paper proposes a new exploration scheme for general multi-agent reinforcement learning called PMIC. PMIC maintains two separate buffers by keeping trajectories with high rewards and trajectories with less satisfying rewards. PMIC additionally computes a lower bound of mutual information between state and policy over the positive buffer and an upper bound of the negative buffer. Finally, an intrinsic reward in the form of the lower bound minus the upper bound is introduced as the exploration bonus, which improves a collection of different MARL algorithms on a few benchmark testbeds.

**Summary Of The Review:**

Post-rebuttal:
After carefully re-checking the paper with the feedback from the authors, I have fully convinced by the methods and therefore changed my recommendation to acceptance.

==================================

The paper proposes a seemingly neat algorithm and presents a good amount of experiments. However, some critical correctness flaws cannot be rigorously justified based on the current content of the paper due to too many hand-wavy arguments. The readability of the paper should be improved as well.

I cannot recommend acceptance based on the current form of the paper. But, I would still believe in the potential of the algorithm based on the experiment results. That being said, the paper still has the potential to become a really strong one if the algorithm can be rigorously (or mathematically) justified.

---

> ### Author Response · Authors · 2021-11-18
> **To Reviewer vXVk - Part 1: Questions and Discussions on Mothodology and Others (1)**
>
>
> We appreciate your responsible comments.
> According to your comments,
> we rewrite section 3.1 to help the readers better understand our motivation. We explain motivation more detail in the new version about why simply maximizing mutual information (i.e., the correlation of agents’ behaviors) may prevent the agents from learning better collaboration. Based on your suggestions, we have described in detail what kind of collaboration should be avoided and what kind of collaboration should be enhanced.
> Then we show the experimental evidence to prove this.
>
> We address the reviewer's concerns point by point in the following:
>
>
> (1) [Re: "why does maximizing MI hinder learning?"]
>
> Maximizing the collaboration of agents' behaviours on suboptimal collaboration patterns may lead agents into suboptimal. For example, two agents catch two targets A and B, the team reward table of capturing different targets is shown below:
>
> |         | Agent 1 catches A   |  Agent 1 catches B  |
> | --------   | -----:  | :----:  |
> | **Agent 2 catches A**       |  11   |   -5     |
> | **Agent 2 catches B**        |   3  |   7   |
>
> For example, when agents maximize the correlation of agents' behaviours on suboptimal pattern of capturing B together.
> From agent 1's perspective, agent 1 will improve the certainty that agent 2 is going to capture B; and it is the same from agent 2's perspective. This means that for agents 1/2, after being more certain that the other agent will take action to capture B, the best action of agent 1/2 is to capture B. Thus the suboptimal collaboration will be enhanced. Besides, the experimental results in the paper of VM3[1] also provide some evidence, the performance of VM3 gradually decreases by increasing the weighting factor of MI among agents' behaviors (see (c) and (d) of Figure 4 in VM3's paper). This also supports our motivation that simply maximising mutual information may affect learning to better collaboration.Therefore, it can be important to reduce the correlation between agents on sub-optimal collaboration to break the collaboration and encourages exploration.
>
>
> (2) [Re: "why does the proposed PMIC algorithm solves the issue in this objective"]
>
> The overall motivation can be described as: Instead of enhancing all collaboration patterns, we would like to
> maximize the mutual information of superior collaboration patterns to strengthen them and minimize the mutual information of inferior collaboration patterns to avoid being stuck into these patterns.
> The key point here is to train MINE and CLUB based on two separate buffers, which dynamically store the superior patterns the inferior patterns separately based on rewards. Then we leverage MINE and CLUB trained on the two buffers.
> Finally PMIC can provide the MI signals based on MINE and CLUB to any collaboration to determine how far to punish or strengthen this pattern.
>
> (3) [Re: “What is the relationship between Eq.(1) and PMIC”]
>
> We have discussed the relationship of PMIC and Eq.(1) in section 3.3. Here we try to provide a more detailed and helpful explanation. One thing to note is that in the motivation section we present some of the disadvantages of previous mutual information calculations for improving the correlation of agents' behaviors.
> To the end, we propose a new way to measure collaboration pattern $I(s;\pi(u|s))$ without introducing additional variables and suffering from the scalability issue. We demonstrate the performance benefits of $I(s;\pi(u|s))$ in Figure 14 and efficient guidance in Figure 9.
> As we discussed above, blindly maximizing MI may enhance correlation of agents on inferior collaboration, which limits the performance of the overall algorithm. We therefore design PMIC which leverages our proposed $I(s;\pi(u|s))$ as a measure of collaboration.
>
>
> (4) [Re: “connection between rewards and MI”]
>
> MINE/CLUB provides an estimate of the bound of MI, which can be expressed as a quantitative measure of whether any other pattern fits these two patterns, i.e., superior collaboration patterns and inferior collaboration patterns.
> Combined with our main idea, i.e., progressiveness, we leverage this quantification as the intrinsic reward to achieve our goal.
>
> [1] Woojun Kim, Whiyoung Jung, Myungsik Cho, and Youngchul Sung. A maximum mutual information framework for multi-agent reinforcement learning. arXiv preprint arXiv:2006.02732, 2020.

---

> ### Author Response · Authors · 2021-11-18
> **To Reviewer vXVk - Part 2: Questions and Discussions on Mothodology and Others (2)**
>
>
> (5) [Re: “Why only the behaviors that obey superior collaboration patterns or inferior collaboration patterns have large MI estimates calculated by MINE or CLUB?”]
>
> Let's first explain how MINE/CLUB works. The mutual information of any two variables can be defined as:
>
> \begin{equation}
> I(X, Z)=D_{K L}\left(\mathbb{P}\_{X Z} \| \mathbb{P}\_{X} \otimes \mathbb{P}\_{Z}\right),                           (1)
> \end{equation}
> where $D_{K L}$ is defined as:
>
> \begin{equation}
> D_{K L}(\mathbb{P} \| \mathbb{Q}):=\mathbb{E}\_{\mathbb{P}}\left[\log \frac{d \mathbb{P}}{d \mathbb{Q}}\right].                 (2)
> \end{equation}
>
> The meaning of equation (1) is clear: the larger the divergence between the joint and the product of the marginals, the stronger the dependence between X and Z. This divergence, hence the mutual information, vanishes for fully independent variables[2].
> Instead of using KL divergence, we leverage
> the lower bound of our MI based on Jensen Shannon divergence[3] as follow:
>
> \begin{equation}
> I(s;\pi(\cdot|s)) \ge I_{\text{MINE}}(s;\pi(\cdot|s)) = \sup_{\omega_1 \in \Omega}
> \underbrace{\mathbb{E}\_{\mathbb{P}\_{\mathcal{S} \mathcal{U}}} \left[-sp\left(-T_{\omega_1}(s, u)\right)\right]}_{term 1}  -
> \underbrace{\mathbb{E}\_{\mathbb{P}\_{\mathcal{S}} \otimes \mathbb{P}\_{\mathcal{U}}} \left[ sp\left(T\_{\omega_1}(s, u)\right) \right]}\_{term 2},(3)
> \end{equation}
>
> According the above equation, MINE fits mutual information to maximize the divergence between estimates from $\mathbb{P}\_{\mathcal{S} \mathcal{U}}$ and $\mathbb{P}\_{\mathcal{S}} \otimes \mathbb{P}\_{\mathcal{U}}$. Specifically, the network output value of term 1 based on the joint will be higher, while term 2 based on the margins will be suppressed[2].
> This means that the patterns in the joint will have high correlation, while the pattern from the margins will have no correlation (i.e.,$\mathbb{P}\_{\mathcal{S} \mathcal{U}} = \mathbb{P}\_{\mathcal{S}} \otimes \mathbb{P}\_{\mathcal{U}}$ ).
> Thus if the joint of new collaboration patterns is more similar with the joint, it will have a high value estimated by MINE trained with the joint. Otherwise a lower value is obtained.
> As to CLUB,
> CLUB firstly train its prediction network based on the joint, which means CLUB has high conditional log-likelihood on the joint. Then CLUB leverages the difference between the conditional log-likelihood of positive sample pairs from the joint and negative pairs from the margins[4] as MI estimates. The patterns in the joint will have higher differences (i.e., upper bound of mutual information) than patterns not in the joint. Thus if we train CLUB based on the inferior joint, CLUB will provide high values to the new joint which is similar to the inferior joint and provide low values to the joint unlike the inferior joint.
>
>
>
> (6) [Re: "why not optimize MINE/CLUB using the entire buffer?"]
>
> As discussed above,
> after training, MINE/CLUB provides an estimate of the bound of MI, which can be expressed as a quantitative measure of whether any other new patterns fit the joint distribution which is used to train MINE/CLUB. Maximizing MI means leading towards the joint and minimizing MI means moving away from the joint.
> Using replay buffer means using the joint distribution of patterns composed of all historical inferior and superior data. Training MINE/CLUB with such a mixed joint distribution can not provide agents effective signals. It is the reason why MINE and CLUB should not be trained based on the entire replay buffer, which violates our motivation and contribution points. Besides, we have provided the experimental result of PMIC-MADDPG with entire replay buffer in Figure 10, which supports our view.
>
>
>
> [2]  Mohamed  Ishmael  Belghazi,  Aristide  Baratin,  Sai  Rajeshwar,  Sherjil  Ozair,  Yoshua  Bengio,Aaron Courville, and Devon Hjelm. Mutual information neural estimation. In International Confer-ence on Machine Learning, pp. 531–540, 2018
>
> [3] Hjelm, R. D., Fedorov, A., Lavoie-Marchildon, S., Grewal, K., Trischler, A., and Bengio, Y. Learning deep
> representations by mutual information estimation and maximization. arXiv preprint arXiv:1808.06670, 2018.
>
> [4] Pengyu Cheng, Weituo Hao, Shuyang Dai, Jiachang Liu, Zhe Gan, and Lawrence Carin.  Club:A contrastive log-ratio upper bound of mutual information. In International Conference on MachineLearning, pp. 1779–1788. PMLR, 2020.

---

> ### Author Response · Authors · 2021-11-18
> **To Reviewer vXVk - Part 3: Questions and Discussions on Mothodology and Others (3)**
>
> (7) [Re: "the joint policy should be diverse enough to avoid being stuck into sub-optimal collaboration."]
>
> In fact, the word "diverse" means the action variable decided by joint policy should be diverse (since $\pi$ is the distribution of action).
> This is because if the action variable collapses to the same single value for all states in some extreme case, the joint policy is fully deterministic but the "collaboration" in this case is spurious and makes no sense.
> Maximizing $H(\pi(·|s))$ encourages agents to explore in the joint action space to enrich the diversity of the collaboration pattern.
> We appreciate the reviewer for pointing out this and we have clarified this to make it clearer in the revised version.
>
>
> (8) [Re: "The predator-prey environment is not a cooperative one"]
>
> The predator prey is originally a competitive environment. In our setting, we consider training the algorithms to control the predators for collaboration. The policy of prey is fixed and identical for different algorithms. We have clarified this to make it clearer in the revised version.
>
> (9) [Re: "I don't get why this point is necessary when we have 2) already."]
>
>
> The second term does not model the dependence among collaborative agents.
> Considering practical collaboration problems where there are multiple collaboration patterns given the global state, maximizing $-H(\pi_i\mid s)$ may encourage agents' policy to concentrate its probability mass on several collaboration patterns.
> For the third term,
> maximizing $-H(\pi_{-i}\mid\pi_{i},s)$ reduces the uncertainty of the agent about the policy of other agents.
> It models the dependence among agents, i.e., the information how other agents are likely to behave if one chooses a specific choice of collaboration pattern.
> Therefore, the third term is necessary in this sense.
>
>
> (10) [Re: "A possibly related paper"]
>
> We thank the reviewer for bringing the related work into our attention. It definitely has some connections to our work. We added a reference to it in the updated version of the paper.
>
>
> We would really appreciate it if you can let us know **if you have any pending concerns and if our response addressed some/all of your concerns**. **We will try to address them before the discussion period ends**.

---

> ### Author Response · Authors · 2021-11-26
> **Response to POST REBUTTAL Comments from Reviewer vXVk**
>
>
> We are pleased to have addressed your concerns and misunderstanding!
>
> We appreciate your suggestions very much.
> We have added Related Work in Appendix A and we will streamline our main text and move the Related Work back to the main text of our paper. Moreover, we will re-consider the organization of the content in section 3.1 and 3.2
> to make the presentation more consistent for the readers to follow. Careful amendment on the valuable suggestions will be seriously made and considered in our later revision.

---

### Author Response · Authors · 2021-11-18
**We appreciate all the reviewers' careful and valuable comments**

We appreciate all the reviewers' careful and valuable comments. Individual responses and our revision will be provided soon.

We are looking forward to the following inspiring discussions.

---

### Author Response · Authors · 2021-11-18
**Updates Made in Our Revisions**

We have uploaded a revision in which all modifications are colored in orange.

Major updates are summarized below:



$\bullet$ Related Work is added in the appendix.

$\bullet$ We rewrite Sec.3.1 to give a more clear formulation of the motivation.

$\bullet$ Additional experimental results are provided, including:

$\qquad\circ$ Add the ablation result about PMIC-MADDPG with normal buffer in Figure 10.

$\qquad \circ$ Add ablation study about MINE and normal MI estimation method in Figure 16.

$\qquad\circ$ Add PMIC-MASAC experiments in Figure 19.

$\bullet$ Some other experimental details are added.


Further updates made in following revisions will be informed.

---

### Decision · Program_Chairs · 2022-01-20

**Decision:**

Reject

**Comment:**

This paper includes an interesting idea of pushing towards good, and away from bad, trajectories, in a natural clean way.

The main problem of the paper is one of clarity.  The paper could be written to be more concise and clear, which would allow, for instance, for sufficient space for the figures (which are currently sometimes rather tiny) as well as not having to fiddle with the margins and spaces quite as much as the current submission seems to do (which would be strictly disallowed at most conferences).  The issue of clarity was also clear during discussion, where sometimes multiple rounds of clarifications were needed to allow the reviewers to correctly interpret parts.

For these reasons, I recommend that the authors resubmit a new, cleaned-up, version of the work, with all the changes neatly incorporated.  Then I think this could make for a nice addition to the literature.

I appreciate this will be a disappointment to the authors, but I think ultimately it will make their work more impactful, and longer-lasting.